# Locus-level L1 DNA methylation profiling reveals the epigenetic and transcriptional interplay between L1s and their integration sites

## Graphical abstract

## Authors

Sophie Lanciano, Claude Philippe, Arpita Sarkar, ..., Laure Ferry, Pierre-Antoine Defossez, Gael Cristofari

## Correspondence

gael.cristofari@univ-cotedazur.fr

## In brief

L1 retrotransposons are abundant sequences in the human genome, implicated in disease and evolution. Lanciano, Philippe et al. profile their DNA methylation patterns at single-locus resolution in human cells and investigate the intricate relationship between L1 DNA methylation and expression and the epigenetic and transcriptional landscape of neighboring genomic regions.

## Highlights

- Bs-ATLAS-seq profiles L1 position and methylation genome wide

- L1's influence on upstream methylation is frequent but limited to 300 bp

- Estrogen receptor binds to unmethylated L1, driving L1-derived transcript expression

- L1 hypomethylation alone is insufficient to enable its transcription at most loci

Lanciano et al., 2024, Cell Genomics 4, 100498
February 14, 2024 © 2024 The Authors.

CellPress

## Article

# Locus-level L1 DNA methylation profiling reveals the epigenetic and transcriptional interplay between L1s and their integration sites

Sophie Lanciano,[1,3] Claude Philippe,[1,3] Arpita Sarkar,[1] David Pratella,[1] Cécilia Domrane,[2] Aurélien J. Doucet,[1] Dominic van Essen,[1] Simona Saccani,[1] Laure Ferry,[2] Pierre-Antoine Defossez,[2] and Gael Cristofari[1,4,*]

[1]University Cote d'Azur, INSERM, CNRS, Institute for Research on Cancer and Aging of Nice (IRCAN), Nice, France
[2]University Paris Cité, CNRS, Epigenetics and Cell Fate, Paris, France
[3]These authors contributed equally
[4]Lead contact
*Correspondence: gael.cristofari@univ-cotedazur.fr

## SUMMARY

Long interspersed element 1 (L1) retrotransposons are implicated in human disease and evolution. Their global activity is repressed by DNA methylation, but deciphering the regulation of individual copies has been challenging. Here, we combine short- and long-read sequencing to unveil L1 methylation heterogeneity across cell types, families, and individual loci and elucidate key principles involved. We find that the youngest primate L1 families are specifically hypomethylated in pluripotent stem cells and the placenta but not in most tumors. Locally, intronic L1 methylation is intimately associated with gene transcription. Conversely, the L1 methylation state can propagate to the proximal region up to 300 bp. This phenomenon is accompanied by the binding of specific transcription factors, which drive the expression of L1 and chimeric transcripts. Finally, L1 hypomethylation alone is typically insufficient to trigger L1 expression due to redundant silencing pathways. Our results illuminate the epigenetic and transcriptional interplay between retrotransposons and their host genome.

## INTRODUCTION

Transposable elements represent a considerable fraction of mammalian genomes and contribute substantially to their gene-regulatory networks.[1,2] In humans, the long interspersed element 1 ( L1, also known as LINE-1) retrotransposon accounts for at least 17% of the genome and is the sole autonomously active transposable element.[3] Its dramatic expansion to around half a million copies results from a copy-and-paste mechanism called retrotransposition, orchestrated by L1 mRNA and L1-encoded proteins, ORF1p and ORF2p.[4–7] L1 expression is primarily controlled by a CpG-rich bidirectional promoter located within its 5′ untranslated region (UTR) (Figure 1A). The sense promoter (SP) drives the synthesis of L1 mRNA, with a significant fraction of readthrough into the downstream genomic sequence.[8–12] The antisense promoter (ASP), on the other hand, can function as an alternative promoter for neighboring genes, leading to spliced chimeric transcripts, or even to fusion proteins with an L1-encoded antisense open reading frame (ORF), designated ORF0.[13–20]

The propagation of L1 elements throughout mammalian evolution occurred in successive waves of expansion and extinction implicating a limited number of concurrent families. In anthropoid primates, a single lineage, L1PA, has been active, leading to the sequential emergence of the L1PA8-L1PA1 families

(from the oldest to the youngest) over the past 40 million years.[21] L1PA1, also known as human-specific L1 (L1HS), is the only family capable of retrotransposition in modern humans,[22,23] with an estimated insertion rate of one new heritable insertion every 60 births.[24] Although all L1PA sequences are related, the 5′ UTR and ORF1 regions exhibit rapid adaptive evolution, likely spurred by an arms-race with the host genome.[25–27] Upon integration, L1 sequences accumulate alterations such as mutations, indels, or nested transposable element insertions, further diverging from their original progenitor.[28] Thus, despite being highly repeated, L1 elements exhibit considerable heterogeneity, both within and between families. This is particularly evident in their promoter region, suggesting variable regulatory mechanisms. Consistently, a variety of Krüppel-associated box domain zinc-finger proteins (KZFPs) specifically bind L1PA8 to L1PA3 elements in different cell types, leading to TRIM28-mediated silencing,[27,29,30] whereas silencing of younger L1PA2 and L1HS elements presumably occurs through TRIM28-independent mechanisms, such as DNA methylation.[29,31–36] The human silencing hub (HUSH) complex also represses a subset of young L1 elements in various cell types.[37–39]

Despite these repression mechanisms, L1HS elements can mobilize in the germline and early embryonic development, leading to heritable genetic variations and occasionally causing genetic diseases.[40] These additional insertions, not cataloged in the

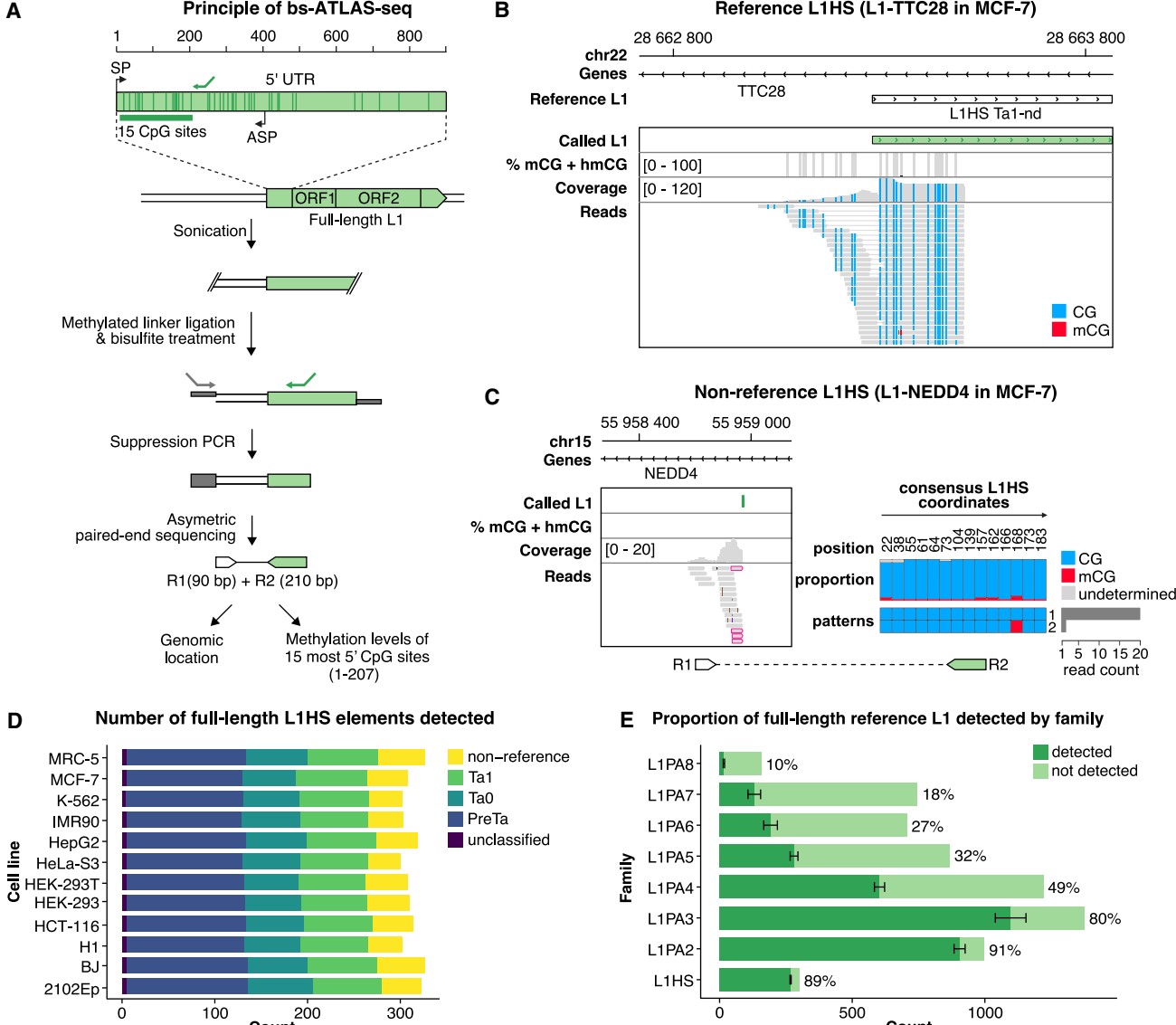

**Figure 1. Bisulfite-ATLAS-seq (bs-ATLAS-seq) profiling of human L1 element promoters**

(A) Principle of bs-ATLAS-seq. L1 junctions encompassing the first 15 CpG sites of the L1 promoter (dark green vertical bars) are amplified using an L1-specific primer (green arrow) designed to target the L1HS family. R1 and R2 refers to read 1 and 2, respectively. SP, sense promoter; ASP, antisense promoter.

(B and C) Genome browser views of reference (B) and non-reference (C) L1HS elements (MCF-7 cells). CpG methylation is indicated by vertical bars (gray for the site, black for methylation percentage). Coverage and read tracks show non-methylated (CG, blue) and methylated (mCG, red) CpG sites. For non-reference L1HS elements (C), only the genomic region covered by R1 is visible. Pink frames highlight soft-clipped reads supporting the 5′ L1 junction. The proportion of mCG at each site and the frequency of the most common methylation patterns deduced from R2 are indicated (right). CpG positions are relative to the L1HS consensus sequence. mCG and hmCG refer to CpG sites with 5-methylcytosine and 5-hydroxymethylcytosine modifications, respectively.

(D) Number of full-length L1HS elements detected by bs-ATLAS-seq and their lineages (Pre-Ta, Ta0, and Ta1, from the oldest to the youngest).

(E) Bars represent the total number of copies in the reference genome (light green) overlaid by the average number of copies detected by bs-ATLAS-seq (dark green, mean ± SD, n = 12 cell lines). The percentage of detected/total elements is indicated on the right. See also Figure S1 and Table S1.

reference genome, are referred to as non-reference elements. They contribute to human genetic diversity, with two individual human genomes differing, on average, at 285 sites with respect to L1 presence or absence.[23,41–46] Additionally, L1HS elements can also retrotranspose in some somatic tissues, such as the brain,[40] and in numerous epithelial tumors.[47,48] Older L1 elements, while unable

to complete a full retrotransposition cycle, can still produce transcripts including long non-coding RNA[49] or act as alternative promoters contributing to oncogene activation in some tumors.[50–52] Nuclear L1 transcripts and transcriptional activity have also been implicated in regulating nuclear architecture, chromatin accessibility, and totipotency in mammals.[53–55]

A prevailing view attributes L1 reactivation in cancer to the global loss of DNA methylation commonly observed in tumor cells.[56] Consistently, targeted analyses of selected progenitor L1 elements confirmed their hypomethylation in tumors compared with normal tissues.[12,22,48,57,58] Similar associations between L1 hypomethylation and activity were found in human pluripotent and neuronal progenitor cells.[33,59–64] However, early studies also hinted at heterogeneity in DNA methylation among different L1 loci, particularly in tumor cells.[31,36] This heterogeneity extends to L1 expression, with only a restricted subset of L1 loci, including a handful of L1HS elements, being robustly expressed in transformed cells[9,22,65] and ultimately acting as source elements for new retrotransposition events in tumors.[12,22,44,48] Altogether, these observations suggest that DNA methylation must be lifted to permit L1 reactivation. However, the extent of L1 DNA hypomethylation in tumor cells and whether DNA demethylation alone is sufficient to promote L1 reactivation remain unclear. Because L1s can efficiently insert into regions with a wide range of chromatin states,[66,67] it is conceivable that the integration site could influence the capacity of a given L1 element to be reactivated upon demethylation. Addressing these questions systematically has been hampered by the technical challenges of assessing L1 DNA methylation genome wide at single-locus resolution, particularly for the most recent L1HS family.[68,69]

Here, we developed bisulfite amplification typing of L1 active subfamilies sequencing (bs-ATLAS-seq) to map individual full-length L1 elements and profile their methylation status. We applied this approach to a panel of normal, embryonal, and tumoral human cell lines. In conjunction with nanopore sequencing and methylation array analyses, this approach enabled us to identify methylation patterns specific to distinct L1 families and human tissues and to study the regulation of individual L1 copies. We observed that L1 methylation and expression can influence the transcriptional and epigenetic landscape of the genomic region in which they reside and vice versa. We also found that most L1 elements cannot be activated solely by reducing DNA methylation, indicating the presence of multiple layers of epigenetic regulation at work on individual loci.

## RESULTS

### Genome-wide and locus-specific human L1 DNA methylation profiling

The L1 promoter contains a CpG island within its first half, which coincides with its SP activity (Figure 1A).[70–72] To measure the DNA methylation levels of individual L1HS promoters, we employed bs-ATLAS-seq (Figure 1A), a modified version of ATLAS-seq, originally devised for L1HS mapping in the human genome.[9,73] Bs-ATLAS-seq detects reference and non-reference L1HS insertions genome wide, providing the location and methylation state of their promoters at single-locus, single-molecule, and single-nucleotide resolutions (Figures 1B, 1C, S1A, and S1B). The amplified region within the L1 5′ UTR encompasses the first 15 CpG dinucleotides, including seven that are essential for L1 regulation.[74] Because the methylation status of this region reflects that of the broader internal promoter (see below), we will

refer to it hereafter as "L1 methylation." Bs-ATLAS-seq necessitates as little as 10–20 million read pairs and is highly reproducible (STAR Methods; Figures S1C–S1E).

We applied bs-ATLAS-seq to a panel of 12 human cell lines, including normal fibroblasts (BJ, foreskin fibroblasts; IMR90 and MRC-5, lung fibroblasts), artificially immortalized and transformed cells (HEK-293, adenovirus-immortalized embryonal kidney cells, and HEK-293T, a derivative of HEK-293 transformed by SV40 large T antigen), cancer cell lines (HepG2, hepatoblastoma; K-562, chronic myeloid leukemia; MCF-7, breast cancer; HeLa-S3, cervical cancer; and HCT-116, colon cancer), as well as cells of embryonal origins (H1, embryonic stem cells; 2102Ep, embryonal carcinoma cells) (Figure S1F; Table S1). We compiled a comprehensive database containing the position and DNA methylation levels of each L1 copy in these cell lines (Table S2; Data S1), accessible through a dedicated web portal (https://L1methdb.ircan.org). On average, we identified 312 full-length L1HS elements in each cell line, of which 42 are non-reference insertions and assumed to be full length (Figure 1D). Additionally, we detected an average of 81 L1HS reference elements with an amplifiable 5′ end but annotated as 3′ truncated due to internal rearrangements (Table S2). We identified approximately 90% of all full-length reference L1HS elements (Figure 1E). Undetected copies may be absent from the assayed samples or may be present but missed by the assay. By calculating the recovery rate of L1HS-PreTa elements, an L1HS lineage considered to be fixed in the human population,[23] we estimated sensitivity at 97.2% ± 1.7% (Figure S1G). As a complementary approach, we verified the presence or absence of the undetected reference L1HS copies in whole-genome sequencing data of four cell lines (K-562, HCT-116, HepG2, and MCF-7) and obtained a sensitivity at 97.6% ± 2.1% (Figure S1H). Finally, all detected non-reference L1HS elements were either identified in previous studies by distinct methods or experimentally validated (STAR Methods; Figure S1I), indicating that false-positive insertions are virtually absent (Table S2). Although bs-ATLAS-seq was designed to profile L1HS elements, a significant proportion of older primate-specific families (L1PA2 to L1PA8) is also amplified (Figure 1E).

To further validate bs-ATLAS-seq, we compared its methylation levels at select L1 copies with those obtained using alternative techniques that do not rely on bisulfite treatment. Methylated DNA immunoprecipitation (MeDIP) followed by qPCR and direct Oxford Nanopore Technologies sequencing (ONT-seq) of methylated CpGs consistently replicated bs-ATLAS-seq (Figures S1J and S1K; Table S3). Additionally, genome-wide bs-ATLAS-seq data were cross-validated using High Chromosome Contact map (Hi-C) combined with MeDIP (Hi-MeDIP) in MCF-7 cells.[75] Finally, nanopore sequencing confirmed that the methylation levels of the first 15 CpGs reflect those of the entire L1 CpG island (Figure S1K). Thus, bs-ATLAS-seq is a cost-effective and accurate method for profiling L1 position and methylation genome wide.

### L1 promoter DNA methylation is cell-type and family specific

At first sight, the global methylation profiles across the 12 cell types are consistent with the prevalent view that L1 DNA

**Figure 2. L1 promoter DNA methylation is cell type and family specific**

(A) Distribution of DNA methylation levels (% mCG + hmCG) obtained by bs-ATLAS-seq by family and cell type. Boxplots represent the median and interquartile range (IQR) ± 1.5 × IQR (whiskers). Outliers beyond the whiskers are plotted individually.

(B and C) Differential methylation between young (L1HS–L1PA3-short) and older (L1PA3-long–L1PA8) L1PA elements in all publicly available GEO Illumina 450K array datasets (two-sided Wilcoxon rank-sum test; young L1PAs, n = 695; old L1PAs, n = 189; Table S5). Each data point represents median values aggregated from related samples, and size indicates the number of samples.

(B) Median difference of β values between the two groups plotted against methylation levels of young L1PAs. The top 20 p value samples are labeled.

(C) Volcano plots showing the median difference of β values between L1PA groups and associated −$\log_{10}$(p values) for each group of samples. The top-10 p value samples are labeled.

(D and E) Left: reprogramming of primary cells into iPSCs (data from GEO:GSE51921 and GEO:GSE76372). Center: differentiation of ESCs and iPSCs into differentiated cells (data from GEO:GSE31848 and GEO:GSE116754). Right: treatment of ESCs or iPSCs with LIF+3i, a cocktail that promotes a naive state rather than a primed state (data from GEO:GSE65214).

(D) Extracted data points from the volcano plot shown in (C).

(E) Boxplots representing the median and IQR ± 1.5 × IQR (whiskers) with overlaid individual data points (two-sided Wilcoxon rank-sum tests; n, number of samples).

See also Figure S2 and Tables S2 and S5.

methylation is relaxed in cancer cells[31,76–78] (Figure S2A). However, a closer examination of the L1 methylation landscape across cell types and L1 families reveals a heterogeneity not apparent in aggregate analyses (Figure 2A). In most cell types, even cancer cells, the youngest L1HS family appears to be hypermethylated compared with older L1 families (Figures 2A and S2B). Within the L1HS family, reference and non-reference copies show similar methylation levels (Figure S2C). CpG sites are progressively mutated into TpG over time owing to spontaneous deamination.[79] As a result, older L1 families have fewer CpG sites compared with younger ones[43,80] (Figure S2D). Nevertheless, the fraction of methylated CpG measured by bs-ATLAS-seq for these fixed elements remains unaffected by the actual number of CpG sites in the L1 sequence because reads are mapped against the reference genome and not a consensus sequence. Thus, L1 families with lower CpG density could be more susceptible to inter- and intra-locus heterogeneity, as observed previously for Alu elements.[81]

In the embryonal carcinoma cells 2102Ep and the chronic myeloid leukemia cells K-562, L1HS elements exhibit marked hypomethylation, but with distinct epigenetic contexts. In 2102Ep cells, hypomethylation is restricted to the young L1 families (L1HS, L1PA2, and partially L1PA3), while older L1s and the overall genome remain highly methylated (Figures 2A and S2E). In contrast, K-562 cells exhibit global hypomethylation, with levels comparable with those observed in HCT-116 cells with a double *DNMT1-DNMT3B* knockout and affecting all L1 families equally (Figures 2A and S2E). These observations prompted us to examine the expression and mutational status of key epigenetic factors in the cell lines of the panel (Figure S2F; Table S4). However, DNA methyltransferases (DNMTs) and other regulators of DNA methylation are well expressed in K-562 and other cell types and lack damaging variants. Thus, a loss of function of *DNMT* genes does not explain the profile of K-562 cells.

Most cell lines, including cancer cells, show higher methylation levels for young L1PAs than for old ones, while cells of embryonic origin (2102Ep and, to a lesser extent, H1) display the opposite pattern (Figures 2A and S2G, top). To expand these findings to a broader spectrum of human conditions, cell types, and tissues, we analyzed publicly available datasets generated using Illumina Infinium Human Methylation 450 (450K) BeadChip arrays, which are widely used for large clinical studies. We identified probes overlapping young (n = 695) and old (n = 189) L1PA elements (STAR Methods; Table S5). Despite the limited number of CpG sites interrogated and potential cross-hybridization among closely related L1 loci, the assay effectively captures the relative hypomethylation of young vs. old L1PAs in 2102Ep (Figure S2G, bottom) and, inversely, the relative hypermethylation of young vs. old L1PAs in normal fibroblasts (BJ, IMR90, and MRC-5) and some cancer cell lines (MCF-7, HeLa-S3, and HepG2) (Figure S2G, bottom). For other cell types, the overall trend was consistent with bs-ATLAS-seq results but not statistically significant. Encouraged by this result, we systematically analyzed 450K methylation array data from GEO (~12,000 samples; Table S5). Consistent with bs-ATLAS-seq results and prior nanopore sequencing analyses,[82] we observed that the methylation of young L1PAs is high and generally similar to or higher than that of old L1PAs in most situations. In contrast, young L1PAs appear to be hypomethylated in pluripotent stem cells (embryonic stem cells [ESCs] and induced pluripotent stem cells [iPSCs]), trophoblast, embryonal carcinoma cells, seminoma, placenta, fetal membranes, and hydatidiform moles, situations related to early embryogenesis, extra-embryonic tissues, or male germline tumors (Figures 2B and 2C). The methylation difference between young and old L1PAs increases when primary cells are reprogrammed into iPSCs (Figures 2D and 2E, left) or when ESCs or iPSCs are treated with a cocktail containing LIF (leukemia inhibitory factor) and 3i (three inhibitors) that reverts conventional primed stem cells to a naive pluripotent state (LIF+3i; Figures 2D and 2E, right). Inversely, the methylation difference between the two groups is reduced when ESCs and iPSCs are differentiated (Figures 2D and 2E, center). We note that the high levels of young L1 DNA methylation in our H1 ESCs, which were grown in medium containing LIF and serum, suggest a primed state, as reported previously.[63,83]

Collectively, these data demonstrate that L1 DNA methylation is cell type and family specific, with young L1PA hypomethylation being an uncommon occurrence at the family level, primarily observed in pluripotent cells, early development, and extra-embryonic tissues. In contrast, young L1PA hypermethylation is the most prevalent scenario, even in tumors, supporting the notion that L1 activity in tumors or normal somatic tissues arises from a small subset of L1 loci that evade epigenetic repression.[9,12,22,48,63,84]

## L1 DNA methylation can be influenced by genic activity

To further investigate L1 methylation heterogeneity and determine whether specific subsets of L1 elements exhibit consistent hypo- or hypermethylation, we compared the methylation levels of the youngest L1 loci (L1HS and L1PA2) across the different cell lines (Figures 3A and S3A). We identified 288 shared L1HS copies (either full length or 3′ truncated), with the majority being highly methylated in most cell types (Figure 3A). Excluding 2102Ep and K-562 cells, only a small subset of 59 L1HS loci shows variable methylation levels, with none being invariably unmethylated. Similar results were obtained when polymorphic insertions were included (Figure S3B).

In cells where L1HS elements are globally unmethylated (2102Ep and K-562), we still observe methylated copies (n = 54 and n = 46, respectively; Figure 3A). To comprehend why these individual loci deviate from others, we compared their genomic environment. In K-562 cells, L1 methylation is higher in gene bodies, especially in expressed genes, compared with intergenic regions (Figures S3C and 3B). Consistently, methylated L1s are enriched in expressed genes relative to non-transcribed genomic compartments (Figures 3B, S3D, and S3E). In contrast, L1 methylation profiles in 2102Ep cells appear to be unaffected by the transcriptional state of the integration locus (Figures 3A, 3B, S3D, and S3E). The body of expressed genes is commonly methylated in human cells.[85,86] Our observations suggest that most L1HS DNA methylation in K-562 cells results from their co-transcription with genes, similar to intragenic CpG islands.[87] We conclude that L1HS elements are not globally targeted by DNA methylation in K-562 cells but that some copies become methylated due to their co-transcription with genes. We also

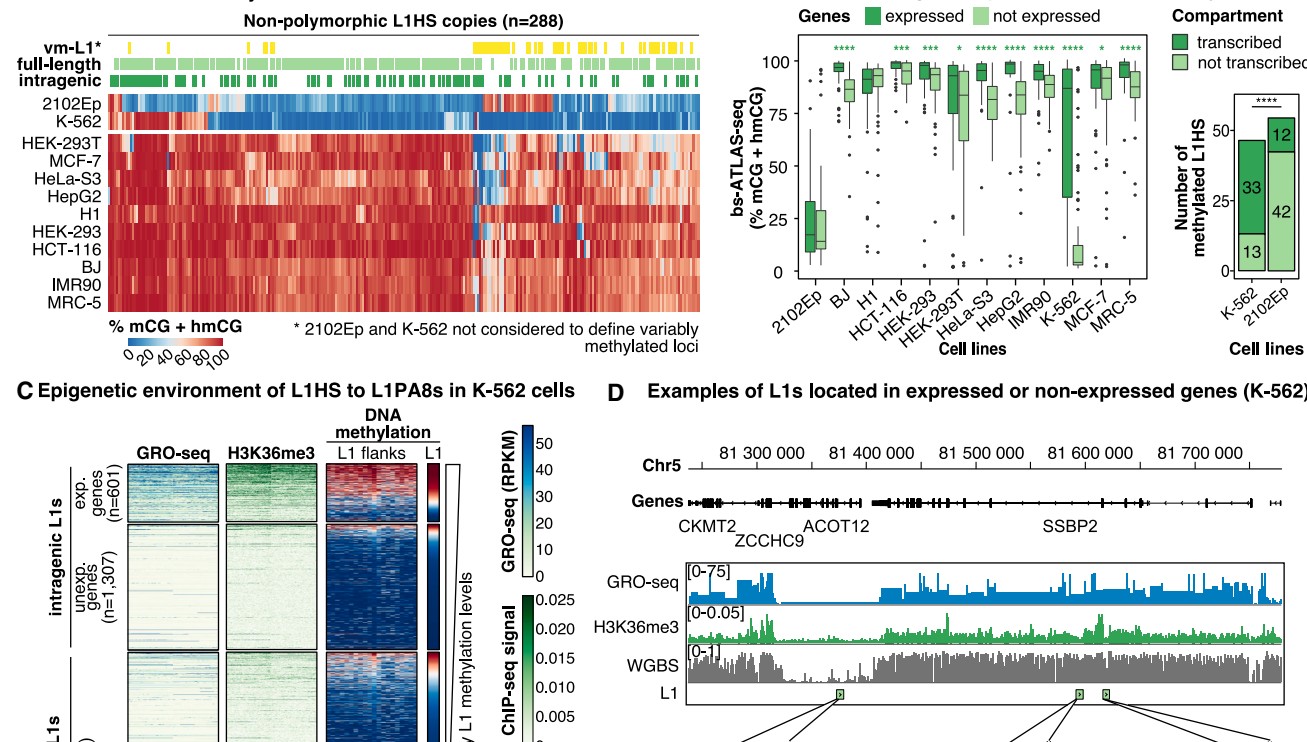

**Figure 3. L1HS promoter DNA methylation is locus specific and influenced by the local environment**

(A) Heatmap of bs-ATLAS-seq methylation levels (% mCG + hmCG) for individual L1HS loci shared by all cell lines. Intragenic (dark green), full-length (light green), and variably methylated (vm-L1s, yellow) L1HS copies are indicated above the heatmap.

(B) Left: comparison of methylation levels for intragenic L1HS elements located within expressed (dark green, transcripts per million [TPM] ≥ 1) or non-expressed genes (light green, TPM < 1). Boxplots represent the median and IQR ± 1.5 × IQR (whiskers). Outliers beyond the whiskers are plotted individually. *p < 0.05, ***p < 0.001, and ****p < 0.0001, two-sided Wilcoxon rank-sum test, with each L1 being considered as an observation. Right: distribution of methylated L1HS elements in transcribed (dark green) and non-transcribed (light green) genomic compartments in K-562 and 2102Ep cells. ****p < 0.0001; chi-square test: $\chi^2(df = 1, N = 100) = 24.6$, p = 7.024e−07.

(C) Heatmaps of nascent transcription (global run-on sequencing, GRO-seq), H3K36me3 ChIP-seq, and DNA methylation (left: whole genome bisulfite sequencing; right: bs-ATLAS-seq) surrounding L1HS–L1PA8 elements (green triangle, ±10 kb).

(D) Genome browser view of L1 elements (green rectangles) with distinct bs-ATLAS-seq methylation profiles (bottom insets), displayed with nascent transcription (GRO-seq), H3K36me3 ChIP-seq, and DNA methylation (whole genome bisulfite sequencing, WGBS) (K-562 cells).

See also Figure S3 and Table S2.

observe a modest association between genic transcription and L1HS DNA methylation in all other cell types except those of embryonic origin (Figure 3B), consistent with previous findings that gene body methylation and expression are not correlated in human ESCs.[85]

Mammalian CpG methylation is directed to gene bodies through the deposition of H3K36me3 during transcription and the subsequent recruitment of the *de novo* DNMT Dnmt3b, at least in mouse early development and germ cells.[85,88–90] Consistently, methylated L1s in K-562 cells are primarily found in transcribed regions with elevated histone 3 lysine 36

trimethylation (H3K36me3) and DNA methylation (Figures 3C, 3D, and S3F). Moreover, the *de novo* DNMTs, DNMT3a and DNMT3b, are expressed in all the cell lines of the panel (Figure S2F). This suggests that a similar mechanism may operate in most cell types, although further investigation is required to identify the specific DNMTs involved.

**L1 elements drive local but short-range epivariation**

Transposable elements have been proposed to function as "methylation centers" from which methylation can propagate into flanking sequences in various species, including

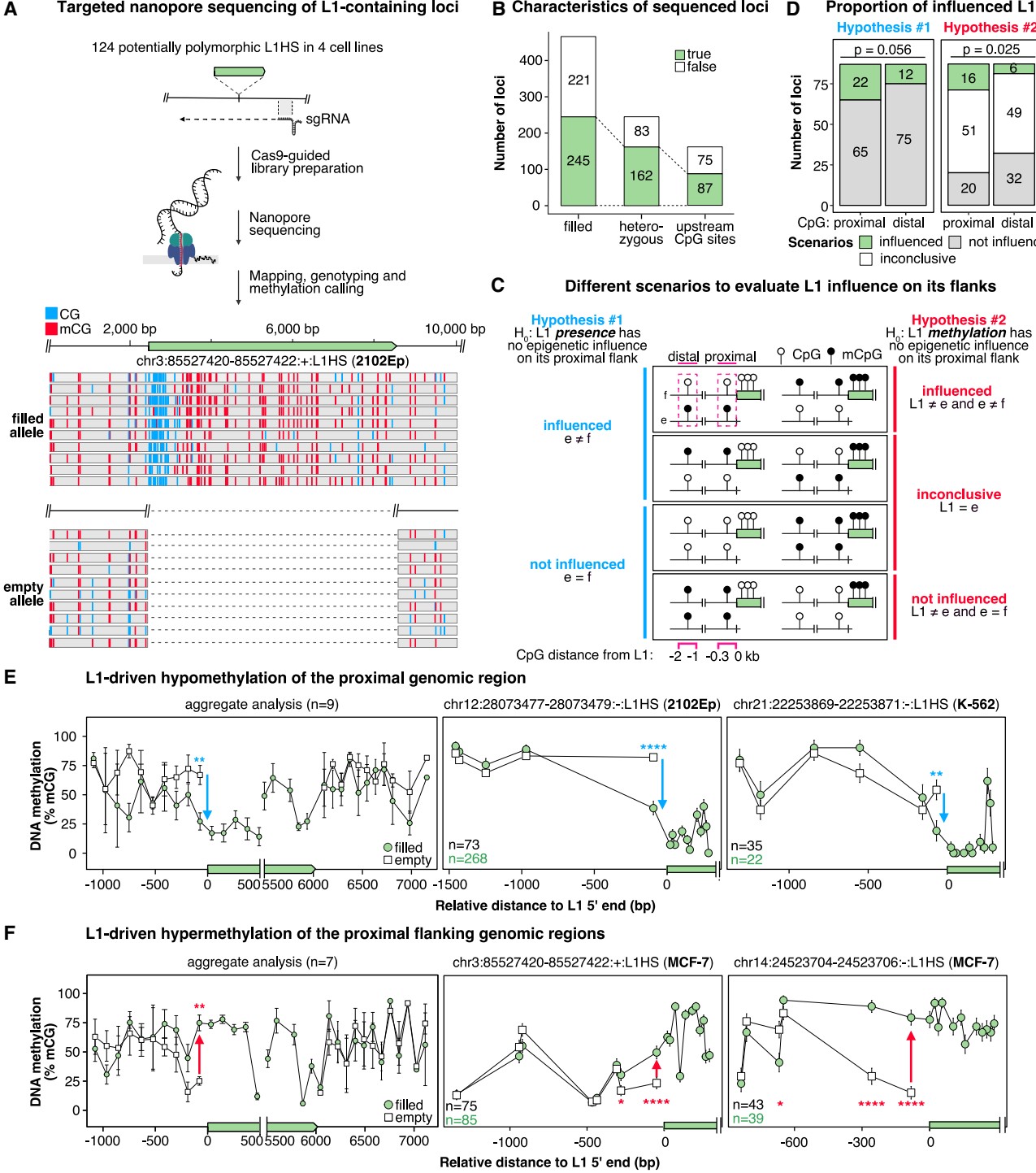

**Figure 4. L1s frequently induce proximal epivariation**

(A) Strategy for genotyping and profiling DNA methylation of L1HS loci by Cas9-guided Nanopore sequencing. Sequencing starts from single-guide RNA (sgRNA) binding sites ~1 kb downstream of target L1s (green) and progresses in the antisense direction relative to L1s. Results are illustrated by a genome browser view of an insertion in 2102Ep cells (top, filled allele; bottom, empty allele). The first 10 reads (truncated at their 3′ end) are shown for each allele with methylated (red) and unmethylated (blue) CpG sites.

(B) Number of L1HS loci sequenced by Nanopore in the 4 cell lines and categorized as filled, heterozygous, and possessing at least one CpG site within a 300-bp upstream window.

*(legend continued on next page)*

mammals.[91] However, this possibility has not been systematically examined for human L1 elements. We found that the DNA methylation status of L1 upstream sequences mirrors that of the L1 promoter itself (Figure S4A). Hypomethylated L1HS and L1PA2 loci show a methylation drop within a 300-bp upstream region, a pattern known as "sloping shore."[92]

To investigate whether L1 elements instruct the methylation state of their flanks, we compared allelic epivariation in the region upstream of L1 at heterozygous loci (i.e., having one filled allele [with L1] and one empty allele [without L1] in the same cell line). To gain access to the genotype and methylation profile of each allele, we employed Cas9-guided Nanopore sequencing with direct calling of 5-methyl-cytosine.[93,94] (Figure 4A). This strategy was applied to 124 potentially polymorphic loci in 4 cell lines (2102Ep, MCF-7, K-562, and HCT-116) with a mean coverage of 45× (Figure 4A; Table S6). On average, 40 loci ± 4 were heterozygous (mean ± SD), and 21 ± 0.5 were homozygous and filled in each cell type, with a total of 162 heterozygous and 83 homozygous filled alleles sequenced (Figure 4B; Table S6). Nanopore sequencing confirmed the methylation levels obtained by bs-ATLAS-seq (Figure S1K) and allowed us to precisely determine the width of sloping shores (Figures S4B and 4A). Methylation starts decreasing approximately 300 bp upstream of L1, reaching a minimum at the beginning of the L1 promoter. Methylation remains low over the first 500 bp of the L1 5′ UTR and increases again to reach a plateau around position +800 with high levels of methylation throughout the entire L1 body, as described previously.[82]

Next, we systematically examined the possibility of L1HS-driven allelic epivariation, considering two hypothetical mechanisms: the presence of L1, regardless of its methylation state, or the methylation state of L1 could influence the methylation of the upstream flanking region (Figure 4C). Among the heterozygous loci, 87 have an upstream CpG within a 300-bp window and were further considered (Figure 4B). To test the first hypothesis (Figure 4C, blue), we compared the methylation levels of the empty and filled alleles at proximal and distal CpG sites upstream of the L1 insertion (Figure 4C, pink dotted frames). While the proximal sites inform us about L1 influence, the distal sites represent the matched genomic background. At proximal sites, 22 of 87 CpGs (25%) exhibit differential methylation between empty and filled loci, but this proportion is not significantly different from that observed at distal sites (Figure 4D, left). To test the second hypothesis (Figure 4C, red), we categorized each locus "influenced" when the upstream CpG is differentially methylated between the empty and filled allele as above and when the average methylation level of the L1 promoter differed by more than 30% from the empty allele. Conversely, a locus

was considered "not influenced" when the upstream region was not differentially methylated between the empty and filled allele, but the average methylation level of the L1 promoter differed from that of the empty allele by more than 30%. Finally, the remaining situations were labeled "inconclusive" (e.g., methylated L1s inserted into methylated regions or the opposite). By comparing the proportion of loci falling in each of these categories, we found that proximal-site methylation is significantly more frequently influenced by L1 methylation than distal-site methylation (Figure 4D, right). We conclude that L1 elements can cause allelic epivariation in their proximal genomic environment by propagating their own methylation state. Almost half of the informative loci exhibit short-range L1-mediated epivariation (16 of 36, 44%). Nine hypomethylated L1 elements have sloping shores (Figure 4E), while 7 hypermethylated elements propagate methylation in their upstream sequence (Figure 4F) and sometimes at both extremities (Figure S4C). For one K-562 locus, we observed broad allelic epivariation between the empty and filled alleles, but it is unclear whether this difference was caused by L1 or rather reflects L1 integration in an existing epiallele (Figures S4D and S4E). Finally, for 20 L1 elements, we did not detect significant variations of methylation between the filled and the empty alleles (Figure S4F). Overall, L1HS elements exert a frequent but short-range epigenetic influence on their genomic environment, creating local epivariation.

### L1-mediated local epivariation is associated with distinct transcription factor binding profiles

To better understand what distinguishes methylated from unmethylated L1HS loci, we compared their transcription factor (TF) binding profiles using Unibind,[95] a curated catalog of chromatin immunoprecipitation sequencing (ChIP-seq) peaks (Figure 5A). For increased statistical power, we also included the L1PA2 family because it represents the most recent primate-specific L1 family after L1HS and shows similar patterns of proximal epivariation (Figure S4A). The screen revealed several TFs specifically associated, in at least one biological condition, with the subset of methylated or unmethylated L1s found in one of the cell lines of the panel (Figure 5B). Closer examination of the motifs underneath the ChIP-seq peaks indicates that some of the binding sites are internal to the L1 promoter (e.g., YY1), and some are found in both the upstream and internal sequences (e.g., ESR1, FOXA1, and CTCF) (Figure 5C).[96,97]

Among the top hits, YY1 is strongly enriched at unmethylated L1s in 2102Ep cells (Figure 5B). This association was detected with Unibind datasets from other cell lines (H1 and NTera2/D1 embryonal carcinoma cells). To corroborate this finding, we conducted YY1 ChIP-seq in 2102Ep cells (Figure 5D) and evaluated

(C) Theoretical scenarios of L1 influence were tested by examining loci with heterozygous L1 insertions (empty allele [e] and filled allele [f]), comparing methylation levels at proximal (300 bp) and distal (1–2 kb) CpG sites (pink dotted frames).
(D) Proportion of the observed scenarios at proximal and distal upstream CpG sites for hypotheses 1 and 2. Chi-square tests of independence for hypothesis 1: $\chi^2$(df = 1, N = 174) = 3.66, p = 0.056, and hypothesis 2: $\chi^2$(df = 2, N = 174) = 7.35, p = 0.025.
(E and F) Allele-specific methylation profiles (empty allele, white squares; filled allele, green circles). Left: average DNA methylation levels in 100-bp bins at loci with (E) an upstream slopping shore (n = 9, mean ± SD) or (F) DNA methylation spreading from L1 to the external flanks (n = 7, mean ± SD). Center and right: examples of L1-induced proximal epivariation. Blue and red arrows denote hypo- and hypermethylation, respectively, relative to the empty locus. Wilcoxon rank-sum tests were used to compare methylation levels between empty and filled alleles, with each locus (left) or each read (center and right) being an observation.
*p < 0.05, **p < 0.01, ***p < 0.001, and ****p < 0.0001; n, number of reads per allele for the filled (green) or empty (black) allele.
See also Figure S4 and Table S6.

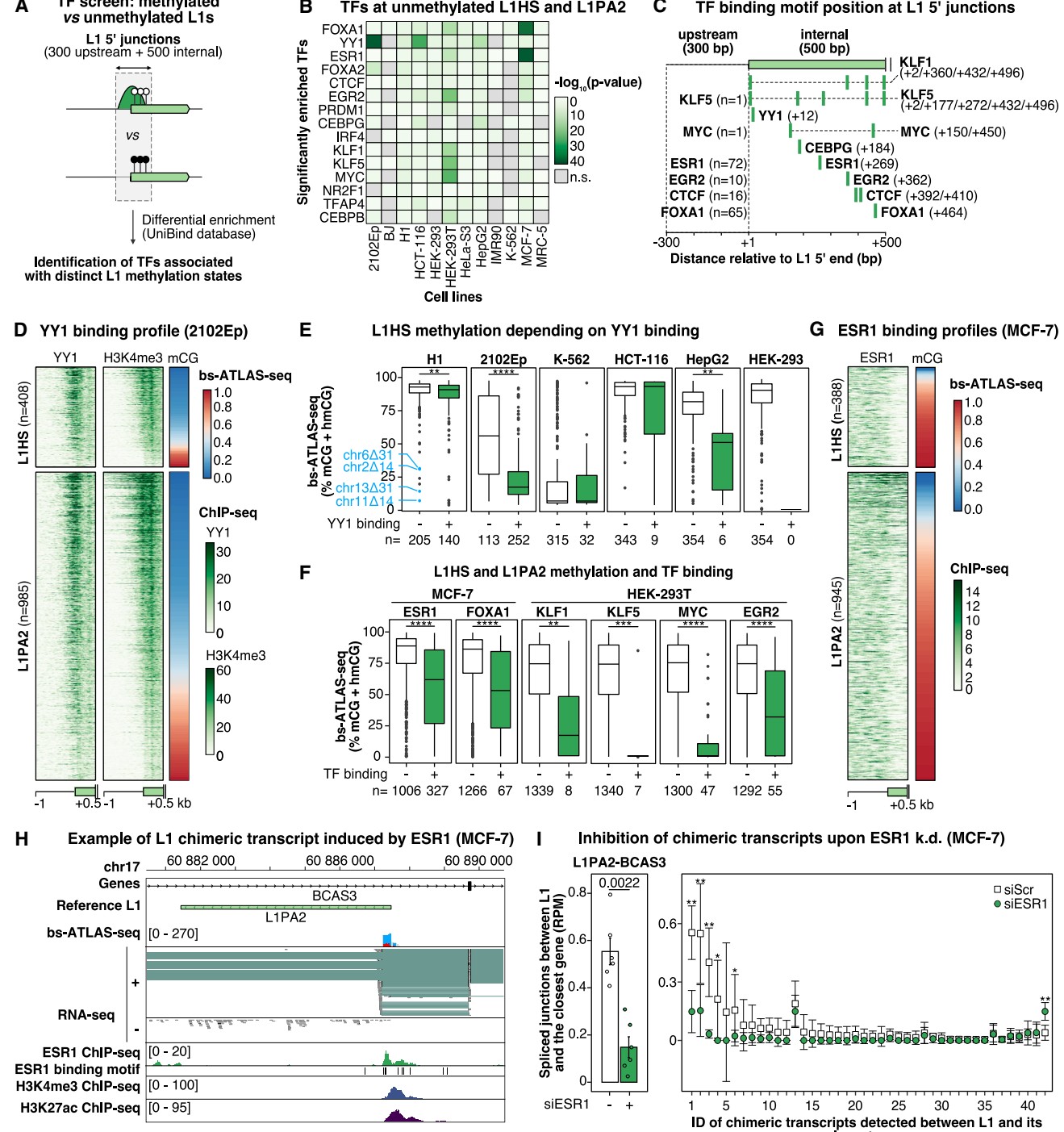

**Figure 5. Unmethylated L1s and their flanking sequences are bound by a specific set of transcription factors (TFs)**

(A) Strategy to identify TFs differentially associated with unmethylated and methylated L1HS and L1PA2 copies using publicly available ChIP-seq data from UniBind[95] (STAR Methods).

(B) Heatmap of TF binding enrichment at hypomethylated L1s, limited to the top 15 hits.

(C) TF binding motifs identified in (B) within L1s and their upstream flank. For TFs binding upstream of L1 insertions, the number of loci with an upstream peak is indicated.

(D) Heatmaps of L1 junctions (2102Ep cells), sorted by L1 methylation (bs-ATLAS-seq) and showing YY1 and H3K4me3 ChIP-seq signals (normalized reads per 10-bp bin).

whether this association was also observed in other cell types of the panel for which YY1 ChIP-seq data were publicly available (Figure 5E). Although YY1 is expressed relatively ubiquitously (Figure S5A), its binding to L1HS elements mostly occurs in the embryonal cell types, H1 and 2102Ep, where YY1-bound elements represent a large fraction of all L1HSs (41% and 69%, respectively), and is limited or negligible in other cell types (Figure 5E). In embryonal cells, YY1-bound elements are significantly less methylated than their unbound counterparts; however, the absolute difference of methylation between the bound and unbound elements is much more prominent in 2102Ep cells than in H1 cells (Δ%mCG = 31% vs. 2%) (Figures 5D and 5E). YY1 also binds a large fraction of L1PA2 elements in H1 and 2102Ep cells, with bound elements being significantly less methylated than unbound ones (Figures 5D and S5B). Focusing on an L1 progenitor with a small 5′ truncation spanning the YY1 motif (named chr13Δ31$_{L1}$) and active in the brain, a previous study proposed that YY1, or another pathway acting on its binding site, may drive L1 methylation.[63] Because YY1 was best known to activate the L1 promoter, helping to define accurate L1 transcription start site,[97–101] this finding was unexpected. We confirmed that the same locus (chr13Δ31$_{L1}$), as well as other L1 elements carrying a similar deletion, are hypomethylated in H1 and 2102Ep ESCs as well as in HepG2 cells (Figure 5E). Therefore, we separately analyzed L1 elements lacking the YY1 motif, possessing the motif but unbound, and those with the motif and bound (Figure S5C). Among unbound L1 elements, those lacking the YY1 motif are less methylated than those with the YY1 motif in H1, HepG2, or HEK-293 cells (Δ%mCG = 21%, Δ%mCG = 17%, and Δ%mCG = 11%, respectively). In 2102Ep cells, YY1-bound L1 elements exhibit the active chromatin mark histone 3 lysine 4 trimethylation (H3K4me3) and higher expression levels compared with unbound elements (Figures 5D, S5D, and S5F). In contrast, the majority of YY1-bound L1HS and L1PA2 copies are hypermethylated in the ESCs H1, and only a minor fraction is hypomethylated and marked by H3K4me3 (Figures S5D–S5F). Considering that 2102Ep cells are nullipotent and blocked in an undifferentiated embryo-like state,[102] and that H1 cells under our growth conditions are likely in a primed state (see above), our observations are thus consistent with a model whereby YY1 preferentially binds unmethylated L1 loci in naive pluripotent cells, thereby enabling accurate L1 transcription initiation, but subsequently mediates L1 *de novo* DNA methylation upon cellular differentiation during development,[63] eventually associated with a loss of YY1 binding. Alternatively, alterations of the YY1 motif may incidentally affect other pathways targeting the same sequence and leading to L1 hypomethylation.[63]

Another prominent hit was the estrogen receptor ESR1, found to be strongly enriched at unmethylated L1 loci in the MCF-7 breast cancer cell line (Figure 5B). ESR1 binds to nearly 25% of L1 loci in these cells, both internally and in their upstream region, with the bound loci being less methylated than unbound ones (Figures 5C, 5F, 5G, and S5G). *ESR1* expression is elevated in MCF-7 compared with other cell types (Figure S5A). Among 327 ESR1-bound L1 elements, 42 form chimeric transcripts with a neighboring gene (Figures 5H and 5I; Table S7). These chimeric transcripts encompass both protein-coding and long non-coding RNAs, many of which are associated with cancer, either as biomarkers or oncogenes, and can encode tumor-specific antigens (e.g., L1-GNGT1).[13,20,50,103] As an example, the promoter of an L1PA2 element residing within an intron of the *BCAS3* gene is unmethylated, marked by active chromatin signatures (H3K4me3 and histone 3 lysine 27 acetylation [H3K27ac]), and bound by ESR1, both internally and in its immediate upstream region (Figure 5H). RNA sequencing (RNA-seq) further shows a high proportion of spliced reads between the L1PA2 ASP and the closest *BCAS3* exon, indicating that L1 hypomethylation and ESR1 binding are associated with ASP activity, which can act as an alternative promoter for *BCAS3*. Unit-length L1 transcripts are also detected at this locus, suggesting that the L1 SP is also active. Upon *ESR1* knockdown using small interfering RNA (siRNA),[104] the expression of L1 elements and their chimeric transcripts diminishes (Figures S5H and 5I). Given the importance of estrogen receptor (ER) status in breast cancer prognosis and management, we explored Pan-Cancer Analysis of Whole Genomes (PCAWG) data to assess whether ER status correlates with increased L1 mobilization in breast cancer.[48,105] We found that ER⁺ tumors more frequently exhibit somatic L1 retrotransposition than ER⁻ tumors (Figure S5I, top). Nevertheless, the number of events is not significantly different between the two groups (Figure S5I, bottom). Consistent with the detection of L1 ORF1p in more than 90% of all breast adenocarcinoma cases, regardless of ER status,[106,107] our results suggest that L1 activation might involve distinct sets of TFs in ER⁻ and ER⁺ tumors. Overall, we conclude that ESR1 directly drives L1 sense and ASP activities in MCF-7 cells and, more broadly, that TFs bound to or adjacent to unmethylated L1s can induce cell-type-specific functional alterations of neighboring genes.

(E) Comparison of DNA methylation levels for YY1-bound (green) or -unbound (white) L1HS elements. N, number of L1HS copies in each subset. In H1 cells, the four hypomethylated loci in blue are those studied by Sánchez-Luque et al.[63]

(F) Comparison of DNA methylation levels of L1HS and L1PA2 elements bound (green) or unbound (white) by various TFs. ChIP-seq data are matched to the cell line. N, number of L1HS copies in each subset.

(G) Heatmap of L1 junctions (MCF-7 cells), sorted by L1 methylation (bs-ATLAS-seq) and showing an ESR1 ChIP-seq signal (normalized reads per 10-bp bin).

(H) Genome browser view of the *BCAS3* locus displaying L1 methylation (bs-ATLAS-seq), expression (poly(A)⁺ RNA-seq), ESR1 binding and H3K4me3/H3K27ac histone modifications (ChIP-seq), and spliced RNA-seq reads linking the L1 antisense promoter with the adjacent *BCAS3* exon.

(I) Differential expression of L1-BCAS3 chimeric transcripts (left) or all 42 identified L1 chimeric transcripts (right) in MCF-7 cells treated with control (−) or *ESR1* (+) siRNAs (data from GEO: GSE153250). Left: bars represent the normalized spliced-RNA-seq read count (mean ± SD, n = 6) overlaid with values from individual replicates. Right: data points represent the normalized spliced-RNA-seq read count (mean ± SD, n = 6) sorted by descending difference between treated vs. untreated cells (see Table S7 for IDs). *p < 0.05, **p < 0.01; two-sided Wilcoxon rank-sum test.

In (E) and (F), boxplots represent the median and IQR ± 1.5 × IQR (whiskers). Outliers beyond the whiskers are plotted individually. *p < 0.05, **p < 0.01, ***p < 0.001, and ****p < 0.0001, two-sided Wilcoxon rank-sum test, with each L1 locus considered as an observation. See also Figure S5 and Table S7.

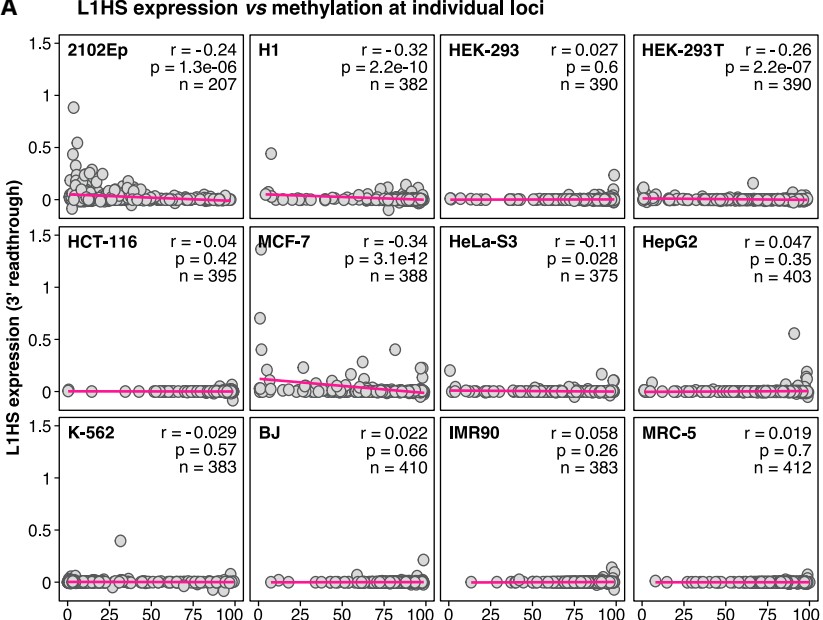

**A** L1HS expression *vs* methylation at individual loci

(Figure A panels: 2102Ep r = -0.24, p = 1.3e-06, n = 207; H1 r = -0.32, p = 2.2e-10, n = 382; HEK-293 r = 0.027, p = 0.6, n = 390; HEK-293T r = -0.26, p = 2.2e-07, n = 390; HCT-116 r = -0.04, p = 0.42, n = 395; MCF-7 r = -0.34, p = 3.1e-12, n = 388; HeLa-S3 r = -0.11, p = 0.028, n = 375; HepG2 r = 0.047, p = 0.35, n = 403; K-562 r = -0.029, p = 0.57, n = 383; BJ r = 0.022, p = 0.66, n = 410; IMR90 r = 0.058, p = 0.26, n = 383; MRC-5 r = 0.019, p = 0.7, n = 412)

Y-axis: L1HS expression (3′ readthrough); X-axis: L1HS methylation (bs-ATLAS-seq)

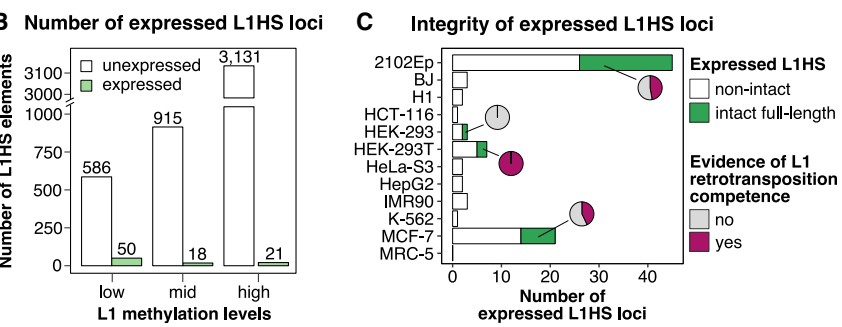

**B** Number of expressed L1HS loci

(Bar chart — unexpressed (white), expressed (green); low: 586 / 50; mid: 915 / 18; high: 3,131 / 21. Y-axis: Number of L1HS elements; X-axis: L1 methylation levels)

**C** Integrity of expressed L1HS loci

(Horizontal bar chart by cell line: 2102Ep, BJ, H1, HCT-116, HEK-293, HEK-293T, HeLa-S3, HepG2, IMR90, K-562, MCF-7, MRC-5. X-axis: Number of expressed L1HS loci. Expressed L1HS: non-intact (white), intact full-length (green). Evidence of L1 retrotransposition competence: no (gray), yes (magenta))

**Figure 6. L1HS promoter hypomethylation is not sufficient to trigger L1HS expression at the locus level**

(A) Correlation between L1HS expression and methylation at individual loci across the different cell lines (r, p, and n represent Pearson's correlation coefficient, p value, and the number of loci, respectively). A regression line is shown in pink.

(B) Number of L1HS elements with low (mCG $\leq$ 25%), medium (25% < mCG < 75%), or high (mCG $\geq$ 75%) methylation levels (bs-ATLAS-seq) in all cell lines combined and identified as unexpressed (white) or expressed (green) by combining the 3′ readthrough (3′ RT) and L1EM[109] approaches.

(C) Number of intact (green) or non-intact (white) expressed L1HS elements. Pie charts show the proportion of intact copies with documented evidence of retrotransposition competence (purple; detailed in Table S8).

See also Figure S6 and Table S8.

Additional TFs found to be enriched at hypomethylated L1s bind to a limited number of L1HS and L1PA2 elements (5% or less; Figure 5F). Among these, FOXA1, highly expressed in MCF-7 cells (Figure S5A), possesses pioneer activity and can drive distance-dependent local demethylation.[108] We thus speculate that FOXA1 could contribute to the cell-type-specific hypomethylation of subsets of L1 loci.

## L1HS promoter hypomethylation is not sufficient to promote expression at most loci

It is broadly accepted that L1 hypomethylation promotes their transcriptional reactivation, at least at the global scale.[33,74] To test this assumption at individual loci, we conducted poly(A)⁺ RNA-seq for the cell lines of the panel because they exhibit a wide range of L1 expression.[9] RNA was prepared from cells collected from the same plate as for bs-ATLAS-seq to match methylation and expression data, except for H1 ESCs, for which publicly available data were used. Measuring L1HS expression levels at individual loci is challenging due to (1) low mappability of repeated sequences, (2) L1HS insertional polymorphisms, and (3) pervasive transcription of L1 embedded in genes that greatly exceeds the autonomous transcription of unit-length L1

elements.[28] To identify autonomous L1 expression driven by the L1 promoter, we applied a previously devised strategy that measures readthrough transcription downstream of reference and non-reference L1HS elements after removing potential signal from gene transcription or pervasive transcription[9] (Table S8). To exclude that some expressed copies could escape detection due to a strong polyadenylation signal at or close to an L1 3′ end, we also assessed L1 expression by L1EM,[109] a software employing the expectation maximization algorithm to reassign internal multimapping reads (Table S8).

Irrespective of the detection method used, only a limited number of L1HS elements are expressed in a given cell type, consistent with our previous findings[9] (Figures 6A and 6B; Table S8). Of note, some of these expressed loci possess intact ORFs and have been documented to be capable of retrotransposition, as evidenced by cell culture assays or by analyzing transduction events originating from these loci (Figure 6C; Table S8). As expected, we observed a weak but significant negative correlation between L1HS methylation and expression in most cell types with detectable levels of L1HS expression (Figure 6A; Table S8). The most highly expressed L1HS elements are unmethylated (Figures 6A and S6A), and fully methylated loci are not expressed (Figures 6A and S6B). However, most unmethylated loci remain unexpressed, indicating that hypomethylation of L1HS elements alone is insufficient to enable their expression (Figures 6A, 6B, and S6C). K-562 cells represent an extreme case of this scenario because only a single element exhibits detectable expression despite most copies being hypomethylated. Conversely, some L1HS copies with relatively high methylation levels are expressed in MCF-7 and HepG2 cells (Figures 6A and S6D). Methylated copies could escape silencing by using an upstream alternative

**A** Differential TE expression analysis (AZA *vs* DMSO)

- up-regulated TE families upon AZA treatment (n=352)
- down-regulated TE families upon AZA treatment (n=6)
- no significant change

**B** Upregulated L1HS loci (AZA)

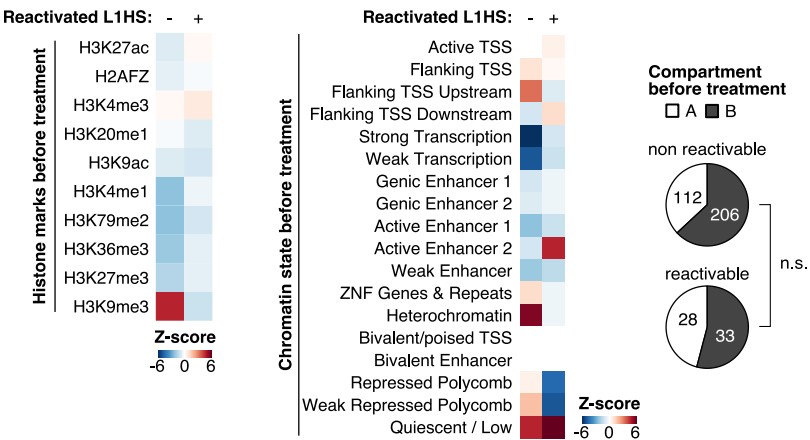

**C** Epigenetic state of AZA-reactivable vs non-reactivable L1HS loci

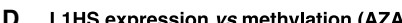

**D** L1HS expression *vs* methylation (AZA)

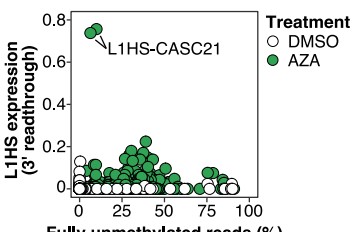

**E** Expression of L1HS-CASC21

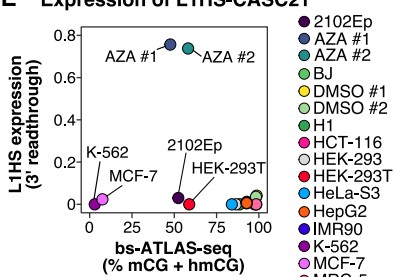

**Figure 7. Acute DNA demethylation only reactivates the expression of a minor fraction of L1HS loci**

(A) Differential expression of transposable element (TE) families between 5-aza- (AZA) or DMSO-treated HCT-116 cells. In the MA plot, each data point represents an aggregated TE family with only L1HS-L1PA8 labeled.

(B) Heatmaps showing the average difference of L1HS methylation (bs-ATLAS-seq, ΔmCG, n = 2) between HCT-116 cells treated with DMSO and 5-aza (AZA) and the expression levels of each L1HS obtained by RNA-seq (3′ readthrough, n = 3). Heatmaps are sorted by decreasing L1 expression (average of the 3 replicates).

(C) Association of reactivable (n = 61) and non-reactivable (n = 318) L1HS elements with pre-treatment histone modifications (left), chromatin states (18-state chromHMM, center), and A/B compartments (right) in HCT-116 cells. Heatmaps show the overlap between L1 flanking sequences (±100 bp) and each genomic feature, expressed as Z scores (blue for depletion, red for enrichment). Pie charts represent the proportion of L1HS elements in A/B compartments for each group (data from ENCODE and Du et al.[113]).

(D) L1HS expression vs. methylation in HCT-116 cells treated (AZA, green) or not (DMSO, white) with 5-aza. Each data point represents an L1HS locus and a replicate.

(E) Comparison of the expression and methylation levels of the intronic L1HS elements inserted into the *CASC21* gene across cell types and conditions.

See also Figure S7.

To experimentally test this hypothesis, we treated HCT-116 cells with 5-aza-2′-deoxycytidine (5-aza, also known as decitabine),[111] a DNMT inhibitor, which homogenously decreases methylation by 50% for all L1HS elements with completely unmethylated reads at most loci (Figures S7A and S7B). As expected, this acute and massive reduction in DNA methylation triggers the activation of several transposable element families, including L1HS–L1PA8 (Figure 7A). However, only a fraction of L1HS copies (16%, 61 of 379 loci) exhibit detectable expression (Figure 7B). 5-Aza-induced demethylation impacts not only promoter re-

promoter.[63,110] Alternatively, epiallele heterogeneity within the cell population or even between alleles might explain their expression because they show a substantial proportion of unmethylated reads (Figure S6D). Besides L1 sense transcription, we also detect ASP activity, with some unmethylated L1HS elements producing only antisense transcripts (Figure S6C). These observations suggest that the absence of DNA methylation at L1HS elements may not always be sufficient to trigger their transcriptional activation.

gions but also gene bodies, with contrasting effects on gene expression,[112] possibly affecting the expression of intragenic L1 elements. To exclude the influence of gene body demethylation on intragenic L1 expression, we separately analyzed L1 loci situated outside of genes, within expressed genes, and within unexpressed genes, combining 3′ readthrough detection and L1EM (Figure S7C). The result shows that the proportion of L1 loci that regain expression after 5-aza treatment is only marginally higher in expressed genes compared with intergenic regions

and that the majority of L1 elements remain silent. Consequently, we conclude that acute DNA demethylation is not sufficient to transcriptionally reactivate most L1 loci.

To understand what differentiates reactivable loci from those that remain repressed, we compared their association with histone modifications and chromatin segmentation states obtained from the Encyclopedia of DNA Elements (ENCODE) project.[114,115] Non-reactivable L1HS loci are enriched in regions with histone 3 lysine 9 trimethylation (H3K9me3) (Figure 7C, left) as well as in heterochromatin (Figure 7C, center). Consistently, non-reactivable elements are proportionally more abundant in B compartments compared with reactivable ones, although the difference is not statistically significant (Figure 7C, right). These observations suggest that multilayers of epigenetic repression coexist within the same cell type, on the same family, and even on the same locus, possibly persisting at most loci even after acute DNA demethylation, at least in HCT-116 colon carcinoma cells. An L1HS element inserted into the intron of *CASC21* (L1-CASC21) displays robust expression following 5-aza treatment in HCT-116 cells (Figures 7D and S7D). Although completely unmethylated in K-562 and MCF-7 cells, it remains unexpressed in these cells (Figure 7E), supporting the notion that cell-type-specific factors are necessary to activate L1HS expression or that alternative epigenetic pathways may supplant DNA methylation in these cell lines.

## DISCUSSION

Understanding the impact and regulation of L1 elements in humans requires genome-wide strategies capable of profiling the DNA methylation of individual copies. Those belonging to the L1HS and other young primate-specific families are especially relevant due to their retrotransposition potential or their contribution to physiological and pathological transcriptomes. Studying these families is notoriously difficult due to their high sequence similarity, an issue exacerbated by the C-to-T conversion employed in bisulfite sequencing protocols. Moreover, individual genomes exhibit substantial variation from the reference genome regarding the presence or absence of L1HS insertions.[28,68] To overcome these obstacles, we developed bs-ATLAS-seq, a method that provides information on the position and methylation state of L1HS, including non-reference insertions, as well as those of many L1PA elements, at single-locus, single-nucleotide, and single-molecule resolutions (Figure 1). Bs-ATLAS-seq offers specific advantages, including excellent cost effectiveness, requiring only 10–20 million reads per sample, and suitability for partially fragmented genomic DNA, commonly encountered in clinical samples. It can be combined with other technologies, such as nanopore long-read sequencing for haplotype-resolved DNA methylation analysis over entire loci[82,94,116] (as illustrated by our allele-specific methylation analysis; Figure 4), or methylation arrays, which remain a platform of choice for large clinical studies of the epigenome (as illustrated in our screen for differential methylation between L1 families; Figure 3).

We comprehensively mapped and characterized the DNA methylation of young full-length L1 elements in a panel of normal, embryonal, and cancerous cell lines, providing one of the most detailed catalogs of L1 DNA methylation in human cells (https://L1methdb.ircan.org). Most of the latter are top-tier ENCODE cell lines, enabling integration with a wealth of publicly available functional genomics data, thus facilitating the exploration of retrotransposon-host genome interactions. We observed that, in most cell types but embryonic cells, the methylation of intragenic L1s is largely influenced by gene expression (Figure 3). Global L1 DNA methylation is a widely used cancer biomarker and a surrogate for measuring global genome methylation levels.[117] Our findings suggest that deconvoluting this global signal could expand its applicability. At the level of individual L1s, particularly those inserted within genes, L1 DNA methylation could serve as an alternative source of DNA-based biomarkers that capture cell-type-specific gene expression. Similarly, at the level of individual L1 families, it could reveal patterns unique to specific cellular states (Figure 2).

Early observations showed that DNA methylation could spread from retroviruses and transposable elements to neighboring regions.[118–121] However, the number of loci driving such epigenetic alterations varies between species and transposable elements, ranging from a single locus for mouse endogenous retroviruses[122,123] to hundreds of loci in plants.[124] Furthermore, DNA methylation typically spreads over a few hundred base pairs but can extend up to several kilobases via plant-specific pathways such as RNA-directed DNA methylation[124,125] and to even longer distances through retrotransposon transcriptional activity.[126] Finally, hypomethylated CpG islands can induce so-called methylation "sloping shores" in their upstream sequence.[63,82,92] The extent and significance of these phenomena has remained uncertain for human L1 elements. Our survey revealed that approximately 20% of informative loci exhibit DNA methylation spreading from a methylated L1 to the adjacent sequence, while 25% show demethylation of the flanking region upstream of hypomethylated L1s (Figure 4). Methylation levels typically follow a descending or ascending gradient starting from the L1 element, suggesting a direct causal relationship between the insertion and proximal epivariation. Although L1 affects nearby DNA methylation only at short distances (typically within 300 bp), epivariation in the zone of influence is associated with differential binding of TFs and can affect the host transcriptome (Figure 5). These findings parallel recent observations in mice indicating that polymorphic endogenous retroviruses and L1s can alter local chromatin accessibility.[127]

One of our initial questions was whether all methylated L1s are repressed and whether, reciprocally, all unmethylated L1s are expressed. We found that the majority of unmethylated L1s remain silent (Figure 6). Consistently, only a fraction of L1s appear to be reactivable upon acute DNA demethylation by a demethylating agent (Figure 7). We conclude that, for most loci, L1 hypomethylation alone is insufficient to induce its expression and that other mechanisms prevent L1 reactivation in the absence of DNA methylation. We uncovered two non-exclusive scenarios. First, L1 silencing pathways can function redundantly and cohabit with DNA methylation at individual loci. Consistently, we observed that L1HS elements not reactivated upon 5-aza treatment are enriched in H3K9me3-bound heterochromatic regions and B compartments before demethylation

(Figure 7C). Deposition of this repressive mark could involve Setdb1 or other histone methyl transferases,[128–131] depending on the cell type, and be tethered by KZFP-TRIM28[132] or the HUSH complex.[37–39] In other cell types, repression could instead involve SIN3A and the local recruitment of histone deacetylases.[133] Incidentally, we note that HUSH-mediated L1 silencing was first discovered through a CRISPR screen in K-562 cells,[37] in which L1PA elements appear to be virtually devoid of DNA methylation (Figure 2). Knocking out any component of the HUSH complex in K-562 cells leads to a massive increase in L1 expression but has more modest effects in other cell types.[37] Second, the expression of cell-type-specific TF binding within or nearby L1, such as ESR1, can be required to switch from an activable but quiescent state to an active state.[134] Accordingly, knocking down ESR1 expression limits L1 expression in the breast cancer cell line MCF-7 (Figure 5).

Retrotransposons and their chimeric transcripts represent a rich source of tumor-specific antigens that can be recognized by infiltrating T cells or CAR-T cells.[103,135–138] Additionally, they can trigger a viral mimicry state, stimulating innate immunity through nucleic acid-sensing pathways.[39,139–143] By modulating retrotransposon expression, epigenetic drugs can thus enhance both specific and innate antitumoral immune responses. Our findings underscore the importance of delineating the precise pathways governing the expression of individual loci. This knowledge will be instrumental in developing tailored drug combinations capable of reactivating L1 elements and L1-derived tumor-specific antigens relevant to immunotherapy while minimizing off-target effects.

### Limitations of the study

We identified a set of TFs associated with hypomethylated L1s. However, the strategy, based on ChIP-seq data, can unambiguously identify only TFs binding near the L1 5′ end, missing those binding L1s more internally. This limitation will be addressed as methods capable of mapping TF binding sites within repeated sequences are becoming more widely available.[75,144–146] We also present data showing association between L1 methylation state and other genomic features. In some cases, we could reasonably infer causality or underlying mechanisms (L1-mediated epivariation, transcription-mediated methylation, or ESR1-driven L1 chimeric transcript synthesis). However, this remains challenging in other cases due to the dynamic nature of DNA methylation throughout development and differentiation, which is difficult to capture using cell lines.

### STAR★METHODS

Detailed methods are provided in the online version of this paper and include the following:

- KEY RESOURCES TABLE
- RESOURCE AVAILABILITY
  - Lead contact
  - Materials availability
  - Data and code availability
- EXPERIMENTAL MODEL AND SUBJECT DETAILS
- METHOD DETAILS
  - Bs-ATLAS-seq
  - Cas9-targeted nanopore sequencing
  - Methylated DNA immunoprecipitation (MeDIP)
  - LUMA
  - Infinium Human Methylation 450K BeadChip analysis
  - RNA-seq
  - 5-Aza-2′-deoxycytidine treatment
  - ChIP-seq
  - Transcription factor enrichment at unmethylated L1
  - Enrichment of genomic features
  - Detection of L1-mediated epivariation
- QUANTIFICATION AND STATISTICAL ANALYSIS

#### SUPPLEMENTAL INFORMATION

#### ACKNOWLEDGMENTS

This work was supported by Agence Nationale de la Recherche (ANR-16-CE12-0020 to G.C. and P.-A.D.; ANR-21-CE12-0001 to G.C. and D.v.E.; ANR-15-IDEX-0001, ANR-11-LABX-0028, and ANR-19-CE12-0032 to G.C.; and ANR-11-LABX-0071 and ANR-18-IDEX-0001 to P.-A.D.), Fondation pour la Recherche Médicale (DEQ20180339170 to G.C.), Institut National Du Cancer (INCa PLBIO 2020-095 to G.C.), Fondation ARC (PGA1/RF20180206807 to P.-A.D.), and other grants to G.C. from the Canceropôle PACA, INCa and the Region Sud (Projet Emergence), INSERM (GOLD Cross-cutting Program on Genomic Variability), and CNRS (GDR 3546). Equipment acquisition for the GenoMed facility was supported by FEDER, Région Sud, Conseil Départemental 06, ITMO Cancer Aviesan (Plan Cancer) and INSERM. We thank the IRCAN Genomics Core Facility (C. Baudoin) for sequencing, the IRCAN Bioinformatics Service (O. Croce and B. Meyer) for computing resources and hosting the L1MethDB web portal, and members of the Cristofari lab for helpful discussions. We are grateful to the ENCODE Consortium for sharing data and J.-L. Garcia-Perez (University of Edinburgh, UK) for providing H1 genomic DNA.

#### AUTHOR CONTRIBUTIONS

G.C. and P.-A.D. conceived the study and secured funding. C.P. developed the bs-ATLAS-seq procedure. S.L. and G.C. designed and conducted the computational analyses. D.P. developed the web interface to interrogate the data. C.P., A.S., C.D., L.F., A.J.D., D.v.E., and S.S. contributed to other experiments. S.L., P.-A.D., and G.C. wrote the manuscript with input from all other authors. G.C. supervised the project.

#### DECLARATION OF INTERESTS

G.C. is an unpaid associate editor of the journal *Mobile DNA* (Springer-Nature).

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

# STAR★METHODS

## KEY RESOURCES TABLE

| REAGENT or RESOURCE | SOURCE | IDENTIFIER |
|---|---|---|
| **Antibodies** | | |
| Rabbit anti-human YY1 | Diagenode | Cat# C15410345; RRID: AB_3083740 |
| anti-H3K4me3 | Abcam | Cat# ab8580; RRID: AB_306649 |
| **Deposited data** | | |
| Human reference genome: NCBI build 38, hg38/GRCh38 | Genome Reference Consortium | https://hgdownload.soe.ucsc.edu/goldenPath/hg38/bigZips/ |
| Annotations: Unified GRCh38 Blacklist regions | ENCODE | ENCODE: ENCFF356LFX |
| Annotations: Repeatmasker track | UCSC Genome Browser | http://genome.ucsc.edu/cgi-bin/hgTables |
| Annotations: L1 annotation track | This study | https://github.com/retrogenomics/bs-ATLAS-seq |
| Annotations: Comprehensive gene annotation (v29) | GENCODE | https://www.gencodegenes.org/human/release_29.html |
| bs-ATLAS-seq (raw data): 12 cell lines | This study | ArrayExpress: E-MTAB-10895 |
| bs-ATLAS-seq (raw data): aza-treated HCT-116 | This study | ArrayExpress: E-MTAB-12240 |
| bs-ATLAS-seq (processed data): Full-length and 3′ truncated L1 mapping and methylation tables | This study | Table S2 |
| bs-ATLAS-seq (processed data): single-molecule methylation patterns of full-length and 3′ truncated L1 | This study | Zenodo: https://doi.org/10.5281/zenodo.7097318 |
| Bs-ATLAS-seq (processed data): database of full-length and 3′ truncated L1 with position, average methylation and single-molecule methylation | This study | Online portal: https://L1methdb.ircan.org |
| RNA-seq: 11 cell lines | This study | ArrayExpress: E-MTAB-12246 |
| RNA-seq: H1 | ENCODE | ENCODE: ENCLB073SSM |
| RNA-seq: 5-aza-treated HCT-116 | This study | ArrayExpress: E-MTAB-12245 |
| WGBS: K-562 cells | ENCODE | ENCODE: ENCFF660IHA |
| ONT-seq: 125 loci, 4 cell lines (HCT-116, 2102Ep, MCF-7, K-562) | This study | ArrayExpress: E-MTAB-12247 |
| ONT-seq (processed data): methylation table | This study | Table S6 |
| Database: euL1db | Mir et al.[45] | http://euL1db.unice.fr |
| Database: Unibind | Puig et al.[95] | https://unibind.uio.no/ |
| ChIP-seq: H3K36me3 (K-562) | NCBI | GEO: GSM1782705 |
| ChIP-seq: H2AFZ (HCT-116) | ENCODE | ENCODE: ENCFF193VYC |
| ChIP-seq: H3K27me3 (HCT-116) | ENCODE | ENCODE: ENCFF294LZM |
| ChIP-seq: H3K4me3 (HCT-116) | ENCODE | ENCODE: ENCFF187LLD |
| ChIP-seq: H3K9ac (HCT-116) | ENCODE | ENCODE: ENCFF724JXS |
| ChIP-seq: H4K20me1 (HCT-116) | ENCODE | ENCODE: ENCFF730VTQ |
| ChIP-seq: H3K27ac (HCT-116) | ENCODE | ENCODE: ENCFF853VVI |
| ChIP-seq: H3K36me3 (HCT-116) | ENCODE | ENCODE: ENCFF528ZNP |
| ChIP-seq: H3K79me2 (HCT-116) | ENCODE | ENCODE: ENCFF104PUB |
| ChIP-seq: H3K9me3 (HCT-116) | ENCODE | ENCODE: ENCFF158YTR |
| HCT-116 chromHMM 18-state model annotations | ENCODE | ENCODE: ENCFF513PJK |
| HCT-116 A/B compartment | NCBI | GEO: GSE158007 |
| GRO-seq (K-562) | NCBI | GEO: GSM4610686 |
| ChIP-seq: YY1 (2102Ep) | This study | ArrayExpress: E-MTAB-12249 |
| ChIP-seq: H3K4me3 (2102Ep) | This study | ArrayExpress: E-MTAB-12249 |
| ChIP-seq: H3K4me3 (H1) | ENCODE | ENCODE: ENCFF156FYC |
| ChIP-seq: H3K4me3 (MCF-7) | ENCODE | ENCODE: ENCFF078BWS |

*(Continued on next page)*

*Continued*

| REAGENT or RESOURCE | SOURCE | IDENTIFIER |
|---|---|---|
| ChIP-seq: H3K27ac (MCF-7) | ENCODE | ENCODE: ENCFF353CZO |
| ChIP-seq: YY1 (H1) | ENCODE/UniBind | ENCODE/UniBind: ENCSR000BKD |
| ChIP-seq: YY1 (K-562) | ENCODE/UniBind | ENCODE/UniBind: ENCSR000BKU |
| ChIP-seq: YY1 (HCT-116) | ENCODE/UniBind | ENCODE/UniBind: ENCSR000BNX |
| ChIP-seq: YY1 (HepG2) | ENCODE/UniBind | ENCODE/UniBind: ENCSR000BNT |
| ChIP-seq: YY1 (HEK-293) | ENCODE/UniBind | ENCODE/UniBind: ENCSR859RAO |
| ChIP-seq: ESR1 (MCF-7) | ENCODE/UniBind | ENCODE/UniBind: ENCFF746RVZ |
| ChIP-seq: FOXA1 (MCF-7) | ENCODE/UniBind | ENCODE/UniBind: ENCFF099YQL |
| ChIP-seq: KLF1 (HEK-293T) | UniBind | UniBind: EXP035894 |
| ChIP-seq: KLF5 (HEK-293T) | UniBind | UniBind: EXP049095 |
| ChIP-seq: Myc (HEK-293T) | UniBind | UniBind: EXP047291 |
| ChIP-seq: EGR2 (HEK-293T) | UniBind | UniBind: EXP035947 |
| RNA-seq: MCF-7 cells treated with ESR1 siRNA | Broome et al.[104] | GEO: GSE153250 |
| WGS: K-562 | ENCODE | ENCODE: ENCLB557IGA |
| WGS: HCT-116 | NCBI | SRA: SAMN19736696 |
| WGS: HepG2 | ENCODE | ENCODE: ENCFF045JFV |
| WGS: MCF-7 | GEO | GEO: GSM3336911 |
| 450K methylation datasets | GEO | Table S5 |
| **Experimental models: Cell Lines** | | |
| Human: 2102Ep | P. W. Andrews | RRID: CVCL_C522 |
| Human: BJ | ATCC | RRID: CVCL_3653 |
| Human: HCT-116 | ECACC | RRID: CVCL_0291 |
| Human: HEK-293 | ECACC | RRID: CVCL_0045 |
| Human: HEK-293T | A. Cimarelli | RRID: CVCL_0063 |
| Human: HeLa S3 | ECACC | RRID: CVCL_0058 |
| Human: HepG2 | ECACC | RRID: CVCL_0027 |
| Human: IMR-90 | ATCC | RRID: CVCL_0347 |
| Human: K-562 | ECACC | RRID: CVCL_0004 |
| Human: MCF-7 | ECACC | RRID: CVCL_0031 |
| Human: MRC-5 | ECACC | RRID: CVCL_2624 |
| **Oligonucleotides** | | |
| Oligonucleotides for bs-ATLAS-seq and PCR validation | This paper | Table S1 |
| Primers for meDIP-qPCR | This paper | Table S3 |
| **Software and algorithms** | | |
| FASTQC | | http://www.bioinformatics.babraham.ac.uk/projects/fastqc |
| Cutadapt (v3.1) | Martin[147] | https://cutadapt.readthedocs.io/en/stable/ |
| Trimmomatic (v0.32) | Bolger et al.[148] | https://github.com/usadellab/Trimmomatic |
| Bowtie2 (v2.4.1) | Langmead et al.[149] | http://bowtie-bio.sourceforge.net/bowtie2/index.shtml |
| Bismark (v0.22.1) | Krueger et al.[150] | https://github.com/FelixKrueger/Bismark |
| Bwa (v0.7.17) | Li et al.[151] | https://github.com/lh3/bwa |
| STAR (v2.7.5c) | Doblin et al.[152] | https://github.com/alexdobin/STAR |
| Minimap2 (v20.2) | Li et al.[153] | https://github.com/lh3/minimap2 |
| Nanopolish (v0.13.2) | Loman et al.[154] | https://github.com/jts/nanopolish |
| BEDTools (v2.29.2) | Quilan et al.[155] | https://github.com/arq5x/bedtools2 |
| Samtools (v1.3) | Danecek[156] | http://www.htslib.org |

*(Continued on next page)*

*Continued*

| REAGENT or RESOURCE | SOURCE | IDENTIFIER |
|---|---|---|
| Methpat (v2.1.0) | Wong et al.[157] | https://bjpop.github.io/methpat/ |
| Seqtk (v1.3) | GitHub | https://github.com/lh3/seqtk |
| GNU parallel (v20200922) | Zenodo | https://doi.org/10.5281/zenodo.1146014 |
| Picard tools (v1.136) | GitHub | https://broadinstitute.github.io/picard/ |
| GATK (v4.1.4.1) | Poplin et al.[158] | https://gatk.broadinstitute.org |
| Variant Effect Predictor (v110) | McLaren et al.[159] | https://www.ensembl.org/info/docs/tools/vep/index.html |
| Integrative Genomics Viewer (IGV, v2.12.3) | Thorvaldsdóttir et al.[160] | http://software.broadinstitute.org/software/igv/ |
| Deeptools (v3.5.1) | Ramirez et al.[161] | https://github.com/deeptools/deepTools |
| MACS2 (v2.2.7.1) | Zhang et al.[162] | http://liulab.dfci.harvard.edu/MACS/Download.html |
| DESeq2 (v1.30.1) | Love et al.[163] | http://www.bioconductor.org/packages/release/bioc/html/DESeq2.html |
| TEtranscripts (v2.2.1) | Jin et al.[164] | https://github.com/mhammell-laboratory/TEtranscripts |
| L1EM (v1.1) | McKerrow and Fenyo[109] | https://github.com/FenyoLab/L1EM |
| MELT (v2.2.2) | Gardner et al.[44] | https://melt.igs.umaryland.edu |
| R (v4.1.2) | | https://www.R-project.org |
| tidyverse (v1.3.1) | CRAN | https://CRAN.R-project.org/package=tidyverse |
| RColorBrewer (v1.1-2) | CRAN | https://cran.r-project.org/web/packages/RColorBrewer/ |
| ggpubr (v0.4.0) | CRAN | https://cran.r-project.org/web/packages/ggpubr |
| Scripts to process raw bs-ATLAS-seq reads | This paper | https://github.com/retrogenomics/bs-ATLAS-seq https://doi.org/10.5281/zenodo.10416341 |

## RESOURCE AVAILABILITY

### Lead contact
Further information and requests for reagents may be directed to and will be fulfilled by the corresponding author, Gael Cristofari (gael.cristofari@univ-cotedazur.fr).

### Materials availability
This study did not generate any new unique reagents or materials to report.

### Data and code availability
- DNA Methylation (bs-ATLAS-seq and ONT-seq), ChIP-seq and RNA-seq data have been deposited at ArrayExpress (www.ebi.ac.uk/arrayexpress) under accession number E-MTAB-10895 (bs-ATLAS-seq, 12 cell lines), E-MTAB-12240 (bs-ATLAS-seq, aza-treated HCT-116 cells), E-MTAB-12247 (ONT-seq, 4 cell lines), E-MTAB-12249 (H3K4me3 and YY1 ChIP-seq of untreated 2102Ep cells), E-MTAB-12246 (RNA-seq, 11 cell lines), and E-MTAB-12245 (RNA-seq, aza-treated HCT-116 cells). Accession numbers are listed in the key resources table. The genomic location and methylation levels of L1 insertions are summarized in Table S2. Single-molecule methylation patterns for each locus have been deposited at Zenodo under https://doi.org/10.5281/zenodo.7097319 and is publicly available as of the date of publication. All L1 methylation datasets can be interactively queried, filtered and downloaded through an unrestricted web portal (https://L1methdb.ircan.org/). This paper also analyzes existing, publicly available data. These accession numbers for the datasets are listed in the key resources table.
- All original code, including a pipeline to call L1 insertions and CpG methylation from bs-ATLAS-seq data, as well as useful annotation files used in the course of this study, has been deposited at Zenodo under DOI: https://doi.org/10.5281/zenodo.7097318 and is publicly available as of the date of publication. DOI is provided in the key resources table.
- Any additional information required to reanalyze the data reported in this paper is available from the lead contact upon request.

## Article

**CellPress**

## EXPERIMENTAL MODEL AND SUBJECT DETAILS

The cell lines used in this study are identical to those previously characterized in[9] and include primary fibroblasts (BJ, IMR90, MRC-5), embryonic stem cells (H1) and cancer or transformed cell lines (HCT-116, K-562, HEK-293, HEK-293T, HeLa-S3, MCF-7, HepG2, and 2102Ep). All cells were directly obtained either from ECACC (distributed by Sigma) or from ATCC (distributed by LGC Standards), apart from 2102Ep cells (a kind gift of P. W. Andrews, University of Sheffield, UK) and HEK-293T (a kind gift of Andrea Cimarelli, ENS-Lyon, France). H1 human embryonic stem cells were not grown in the laboratory for regulatory reasons but genomic DNA of H1 cells grown in presence of LIF and serum was a kind gift of J. L. Garcia-Perez (University of Granada, Spain). Cells were maintained in a tissue culture incubator at 37°C with 5% $CO_2$ and grown in Dulbecco's modified Eagle medium (DMEM), McCoy's 5A (HCT-116) or RPMI 1640 (K-562) containing 4.5 g/L D-Glucose, 110 mg/L Sodium Pyruvate, and supplemented with 10% FBS, 862 mg/mL L-Alanyl-L-Glutamine (Glutamax), 100 U/mL penicillin, and 100 μg/mL streptomycin. Cell cultures tested negative for mycoplasma infection using the MycoAlert Mycoplasma Detection Kit (Lonza). Cell line authenticity was verified by multiplex STR analysis (Eurofins) and comparison with the DSMZ database (https://celldive.dsmz.de/) or with previously published profiles for H1 and 2102Ep cells.[102,165]

## METHOD DETAILS

### Bs-ATLAS-seq

In brief, genomic DNA is fragmented by sonication and ligated to a single-stranded methylated linker. Linker-ligated DNA is then treated with bisulfite and L1-containing fragments are specifically amplified by suppression PCR. In this approach, the linker is single-stranded and possesses the same sequence as the linker-specific primer (not its complementary sequence). Consequently, amplification only occurs upon prior extension from the L1-specific primer and synthesis of the linker complementary sequence. This strategy prevents linker-to-linker amplification. The L1-specific primer was designed to enrich for the L1HS family, but older related L1PA elements are also amplified (see Figure 1E). Finally, asymmetric paired-end sequencing provides the genomic location as well as the methylation levels of each L1 locus. Note that 5-methylcytosine (5mC) and 5-hydroxymethylcytosine (5hmC) are both protected from bisulfite-induced deamination, thus bs-ATLAS-seq cannot discriminate between these two DNA modifications. A practical protocol for bs-ATLAS-seq is provided in[166] and his detailed below.

### DNA extraction

Genomic DNA was prepared with the QiaAmp DNA Blood mini kit (Qiagen) and quantified by fluorometry using the Quant-iT dsDNA HS Assay and a Qubit fluorometer (Thermo Fisher Scientific).

### Mechanical fragmentation, end-repair and A-tailing

Two micrograms of genomic DNA were sonicated for 6 cycles (6 s on, 90 s off) at 4°C with a Bioruptor NGS (Diagenode), generating average fragments of 1 kb. Fragment size was controlled by capillary electrophoresis with the DNA high sensitivity kit and a Bioanalyzer 2100 (Agilent Technologies). DNA ends were repaired using the End-It DNA End-Repair Kit (Epicentre), and A-tailed with Klenow Fragment (3'-to-5' exo-, New Englands Biolabs) following manufacturer's protocol. At each step, DNA was purified with Agencourt AMPure XP beads (Beckman Coulter) using a 1:1 ratio of beads to DNA solution (v/v).

### Linker ligation

Oligonucleotides LOU2493 (with all C methylated) and LOU2494 (Table S1) were mixed in 5 μL of 1x T4 DNA Ligase buffer (50 mM Tris-HCl pH 7.5, 10 mM $MgCl_2$, 1 mM ATP, 10 mM Dithiothreitol; New England Biolabs) at a final concentration of 80 μM each and annealed by heating at 65°C for 15 min, followed by slow cooling down to room temperature. Four hundred nanograms of fragmented genomic DNA were ligated with a 40-fold molar excess of the duplex linker overnight at 16°C in 50 μL of 1x T4 DNA Ligase buffer supplemented with 400 U of T4 DNA Ligase (New England Biolabs). Excess linkers were removed by two successive rounds of purification with Agencourt AMPure XP beads using a 1:1 ratio of beads to DNA solution (v/v). Note that only the single-stranded methylated oligonucleotide LOU2493 is covalently bound to the 5' ends of the genomic DNA fragments.

### Bisulfite conversion

Two hundred and fifty nanograms of linker-ligated genomic DNA were subjected to sodium bisulfite conversion for 210 min at 64°C using the EZ DNA Methylation Kit (Zymo Research) according to the manufacturer's instructions. After clean-up, converted DNA was kept at 4°C for up to 20 h.

### Suppression PCR

L1 5' junctions were amplified in 40 μL-reactions containing 16 ng of converted and linker-ligated genomic DNA, 0.2 μM of primers, 0.2 μM dNTPs, 1.5 mM $MgCl_2$, 0.8 U of Platinum Taq DNA Polymerase in 1X PCR buffer (Invitrogen). A first primer (LOU2565, or LOU2715 to LOU2724) targets the L1-specific region with a 5' extension corresponding to Illumina Rd2 SP and P7 sequences, with a 10-nt index specific to the sample between them. A second primer (LOU2497) targets the linker (identical to Rd1 SP) and possesses a 5' extension corresponding to Illumina P5 sequence. Primer sequences and annotations are provided in Table S1. Amplification was performed under the following cycling conditions: 1 cycle at 95°C for 4 min; followed by 20 cycles at 95°C for 30 s, 53°C for 30 s, and 68°C for 1 min; and a final extension step at 68°C for 7 min. To reduce PCR stochasticity, each sample was amplified in eight parallel 40 μL-reactions and subsequently pooled. In addition, another reaction was performed in the absence of the L1-specific primer to control for the absence of linker-to-linker amplification. The amplified library was cleaned-up from primers and irrelevant

products by double-sided size-selection with Agencourt AMPure XP beads using a 0.55:0.65 ratio of beads to DNA solution (v/v), to reach an average library size of 450 bp. Finally, a last purification was achieved with Agencourt AMPure XP beads using a 1:1 ratio of beads to DNA solution (v/v) to eliminate potential remaining traces of oligonucleotides.

### Sequencing

Libraries were quantified by qPCR with KAPA library quantification kit for Illumina (Roche) and their size range was checked by capillary electrophoresis using with the DNA high sensitivity kit and a Bioanalyzer 2100 (Agilent Technologies). Libraries were diluted to 1 nM and pooled equimolarly. Pooled libraries were paired-end sequenced with a NextSeq 550 system (Illumina) using a high-output kit and 300 cycles and 20% of PhiX DNA spike-in. To gain access to the methylation state of the first 15 CpG in L1 sequence, paired-end sequencing was performed asymmetrically with 90 cycles for read #1 and 210 cycles for read #2.

### Bs-ATLAS-seq primary analysis

Illumina paired-end sequencing reads were processed to locate L1 elements and to call their methylation status, using the script bs-atlas-seq_calling.sh (v1.1, available at https://github.com/retrogenomics/bs-ATLAS-seq), which steps are summarized below. In each read pair, read #1 is 90 bp long and corresponds to the 5′ flanking sequence of L1, while read #2 is 210 bp-long and corresponds to L1 5′ UTR internal sequence.

### Read trimming, mapping, and filtering

We demultiplexed FASTQ files according to their sample-specific barcode using cutadapt (v 3.1, https://github.com/marcelm/cutadapt). We then verified the presence of bs-ATLAS-seq adapters in the reads and trimmed them with cutadapt. Once trimmed, reads #2 were mapped locally against the first 250 bp of L1HS consensus sequence (Repbase Rel. 10.01) using Bismarck (v0.22.1)[150] allowing soft-clipping. Only pairs for which read #2 mapped to the L1HS consensus in the correct orientation were subsequently analyzed (Samtools v1.3).[156] The selected pairs were mapped against hg38 reference human genome using Bismarck in end-to-end mode using the following options: –minins 250 –maxins 1250 –score_min L,-0.6,-0.6. At this stage, mapped read pairs support L1 elements included in the reference genome (reference L1s). To identify non-reference L1 insertions, we extracted reads #1 from unmapped pairs using seqtk (v1.3, https://github.com/lh3/seqtk) and remapped them alone against hg38 with Bismarck in local mode. This read rescue procedure allowed us to identify: (i) discordant pairs when read #1 mapped end-to-end to hg38; and (ii) split read if the 5′ end of read #1 mapped partially to hg38 but its 3′ end mapped to L1HS consensus sequence. We filtered out discordant pairs with read #2 showing more than 4.5% divergence toward L1HS consensus sequence as they correspond to artifactual chimeras formed with old L1 elements. Finally, properly mapped pairs and read #1 singletons were pooled in a single.bam file, and de-duplicated with Picard tools (v1.136, https://broadinstitute.github.io/picard/). As a conservative assumption, we considered read pairs as redundant if their read #1 starts at the same genomic position, since this situation reflects an identical random break site obtained upon sonication.

### L1 calling

We identified reference L1 by intersecting properly mapped pairs with annotated L1 elements in UCSC repeatmasker track[167] using BEDtools.[155] A minimum of 10 non-redundant reads was required to call a reference L1 element. The coordinates of the elements were extracted from UCSC repeatmasker track. To identify non-reference L1 elements, we clustered reads #1 of discordant pairs and split reads less than 100 bp apart, excluding those intersecting with previously identified reference L1s, using BEDtools. A minimum of 10 non-redundant reads, including at least 2 split reads, was required to call a non-reference L1 element. We used the break point of split reads to precisely define the insertion sites at nucleotide resolution (a 2-bp interval spanning the integration point with 0-based coordinates). Finally, candidate L1 elements not in assembled chromosomes (chr1 to chr22, chrX or chrY) or falling in ENCODE Unified GRCh38 Blacklist regions (ENCODE: ENCFF356LFX) were filtered out with BEDtools.

### L1 CpG methylation calling

We called CpG methylation in individual read pair for each reference and non-reference L1 locus, including any covered upstream L1 flanking sequence, using the bismark_methylation_extractor script from Bismarck. CpG methylation patterns for individual loci were summarized and visualized using MethPat.[157]

### Assessment of bs-ATLAS-seq recovery rate and sensitivity

To estimate the fraction of elements detected by bs-ATLAS-seq in each L1 family, we compiled a list of reference L1 elements using hg38 UCSC repeatmasker track filtered to keep only the assembled chromosomes (chr1 to chr22, chrX and chrY) and to remove elements in ENCODE Unified GRCh38 Blacklist regions. The recovery rate was calculated for each sample, taking into account the sex of the donor (presence or absence of a Y chromosome). Reference L1 elements were considered as full-length if their length is > 5900 bp. Non-reference L1s were assumed to be full-length. L1HS subfamilies (lineages) were deduced from diagnostic SNPs in the reference sequence.[168] Non-reference insertion lineage is unknown due to limited information on L1 internal sequence. Given the ongoing activity of L1HS elements in modern humans, and the fact that the reference human genome is a composite assembly obtained from a small number of individuals, it is expected that a given sample only contains a fraction of reference L1HS elements. Thus, to estimate the sensitivity of bs-ATLAS-seq, we calculated the proportion of recovered L1HS-PreTa, an L1HS lineage fixed in the human population.[23] In a complementary approach, we also checked whether reference L1HS elements not detected by bs-ATLAS-seq were present or absent in a given cell line using publicly available whole genome sequencing data using bwa mem with default parameters to align the data and MELT-deletion tool to detect the absence of reference L1 elements in the sample.[44,151] Finally, empty alleles were manually verified in IGV genome browser.

### Assessment of bs-ATLAS-seq false positive rate

To estimate the percentage of false positive L1 detected by bs-ATLAS-seq, we compared candidate L1 elements with databases of known non-reference insertions (KNR), such as euL1db,[45] the 1000 Genomes Project (1KGP)[23] or previous mapping of L1HS in the same cell lines using 3′ junction amplification and Ion Torrent-based single-end sequencing (3′-ATLAS-seq).[9] Only 3 candidate non-reference insertions appeared unknown (Table S2). chr18:15193133-15193135:-:L1HS:NONREF was validated by nanopore sequencing (Table S6). The two others, chr7:140709367-140709369:-:L1HS:NONREF and chr10:38190899-38190901:+: L1HS:NONREF, were validated by PCR of their junctions with the flanking sequence (Table S1 and Figure S1G).

### Methylation heatmaps

Heatmaps were generated from averaged L1 methylation values over the 15 CpG sites analyzed by bs-ATLAS-seq using the pheatmap and dendsort R packages. Rows and columns were ordered by hierarchical clustering based on methylation values using Euclidean distances and the ward.D2 method. Variably methylated L1 elements were defined as showing a greater difference than 30% between the second highest and the second lowest values of DNA methylation at this locus, excluding K-562 and 2102Ep values.

### Cas9-targeted nanopore sequencing

To sequence polymorphic L1 loci, we performed multiplexed Cas9-targeted nanopore sequencing as described in[94] for 125 genomic regions (124 potentially polymorphic loci and 1 control locus). The loci were selected as present in less than half of the cell lines of the panel, increasing the likelihood of heterozygosity.

### Extraction of genomic DNA

High molecular weight genomic DNA was extracted from freshly pelleted cells using the Monarch Genomic DNA Purification kit (New England Biolabs). Immediately after extraction, DNA was quantified by fluorometry using a Qubit fluorometer and the dsDNA HS Assay kit (Thermo Fisher Scientific). Fragment length (>10kb) was verified by resolving 100 ng of DNA on a 0.8% agarose gel. DNA was stored at 4°C until library preparation, usually the following day.

### Design and synthesis of single guide RNAs (sgRNAs)

We designed one sgRNA for each of the 124 potentially polymorphic L1HS loci (i.e., empty in at least 50% of the cell lines of the panel as determined by bs-ATLAS-seq). Using precomputed SpCas9 sgRNA target prediction and scoring by the CRISPOR tool[169] available in the 'CRISPR Targets' track of the UCSC Genome Browser, we selected sgRNAs in the region 900 to 1,500 bp downstream of the targeted L1s, and with the highest scores (at least 55 for the MIT specificity score[170] and 35 for Moreno-Mateos (MM) efficiency score[171]). A control sgRNA (LOU3161) targeting a unique site on chromosome 9 was included as a positive control. The 125 sgRNAs were synthesized as a pool using the EnGen sgRNA Synthesis kit (New England Biolabs), and purified with the Monarch RNA Cleanup kit (New England Biolabs). The sgRNA pool was quantified with the Qubit RNA Assay kit (Thermo Fisher Scientific), aliquoted and stored at −80°C.

### Library preparation

Cas9 ribonucleoprotein particles (RNPs) were assembled by mixing 60 μmol of the sgRNA pool and Alt-R *S. pyogenes* HiFi Cas9 nuclease V3 (IDT) in equimolar amounts in 30 μL of 1X CutSmart Buffer (New England Biolabs) to reach a final concentration of 2 μM. After a 30-min incubation step at 25°C, RNPs were kept on ice. For each cell line, 5 μg of genomic DNA was dephosphorylated by 3 μL of Quick Calf Intestinal Phosphatase (CIP, New England Biolabs) in a total volume of 30 μL for 10 min at 37°C. Then CIP was inactivated by heating the reaction at 80°C for 2 min. Cas9-mediated cut and A-tailing was achieved by adding 10 μL of the Cas9 RNP pool, 1 μL of 10 mM dATP (Thermo Fisher Scientific) and 1 μL of Taq Polymerase (5 U/μL, New England Biolabs) to the CIP reaction. Reactions were incubated at 37°C for 1 h, at 72°C for 5 min, and then kept at 4°C. As a quality control, we performed qPCR using 1 μL of the reaction saved before and after the incubation step to quantify the relative copy number of the intact *RASEF* locus (the target of sgRNA LOU3161) using a pair of primers flanking the cut site (LOU3322: TCACAGGTTGCACACTGGAA, and LOU3323: AGCTCAGCCACTTTTCAGCT) and a pair of primers in *Sox2* as loading control (LOU0695: CATGGGTTCGGTGGTCAAGT, and LOU0696: TGCTGATCATGTCCCGGAGGT). Cleavage was considered as successful if the number of intact target sites decreased by ∼10 to 15-fold. Then, sequencing adapters were ligated to the digested products using the Ligation Sequencing kit (SQK-LSK-110, Oxford Nanopore Technologies) in reactions containing 40 μL of sample, 20 μL of Ligation buffer LNB, 10 μL of NEBNext Quick T4 DNA Ligase (New England Biolabs), 5 μL of Adapter mix AMX-F and 3 μL of nuclease-free water. After 10 min incubation at 20°C, DNA was cleaned up using AMpure XP beads (Beckman Coulter) with a beads-to-sample ratio of 0.3:1 (v/v), washed with the long-fragment buffer (LFB) to retain fragments ≥ 3 kb, and eluted from the beads in 13 μL of elution buffer (EB) at room temperature for 30 min to further enrich for fragments longer than 30 kb. The purified eluate (∼12 μL with 40–45 fmol of DNA) was ready for sequencing on a MinION flow cell and was kept at 4°C until loading.

### Sequencing of DNA library

A MinION flow cell (R9.4.1, Oxford Nanopore Technologies) was loaded on a Mk1B sequencer and primed with a mix of Flush buffer FB and Flush Tether FLT. Then 75 μL of the DNA library (12 μL of eluate, 37.5 μL of Sequencing buffer SBII and 25.5 μL of Loading beads LBII) were loaded into the flow cell and sequenced for 72 h with the MinKNOW interface (v20.10.6). Base-calling was performed during the sequencing run using Guppy (v4.2.3).

**Cell Genomics**
*Article*

### Bioinformatic analysis

To map reads obtained by the protocol described above, we prepared a custom genome including the two possible alleles (empty or filled) for each target locus. Both alleles contained 50 kb upstream and 1 kb downstream of L1, extracted from the human reference genome hg38. L1 insertion sites were deduced from bs-ATLAS-seq experiments. If the targeted L1 is present in the reference genome, the empty allele was made by removing the L1 sequence with bedtools maskfasta (v2.3). If the targeted L1 was absent from the reference genome, the filled allele was built using *ref*orm (https://github.com/gencorefacility/reform) by introducing an L1 consensus sequence at the insertion point. Thus, the custom genome comprises 250 sequences concatenated in a multifasta file. After indexation, nanopore reads were mapped to the custom genome with minimap2 (v20.2) using the following options: -a -x map-ont.[153] Reads with a mapping quality score (MAPQ) of minimum 20 were sorted and filtered using samtools (samtools view -b -q 20). As reads partially spanning an L1 element without reaching the upstream flank tend to be soft-clipped and to be wrongly mapped, we kept only reads longer than 7 kb. Zygosity was evaluated by calculating the coverage of each allele with bedtools coverage (v2.3). If a single allele (filled or empty) was covered, the locus was considered as homozygous. Inversely, if both alleles were covered, the locus was considered as heterozygous. For each covered allele, methylation calling was performed with nanopolish (v0.13.2).[154] We considered only CpG covered by at least 5 reads. Alignments and methylation were visualized with IGV genome browser (v2.12.3).[160]

### Methylated DNA immunoprecipitation (MeDIP)

MeDIP was performed using the Auto MeDIP Kit on an automated platform SX-8G IP–Star Compact (Diagenode). Briefly, 2.5 μg of DNA was sheared using a Bioruptor Pico to approximately 500-bp fragments, as assessed with D5000 ScreenTape (Agilent). Cycle conditions were as follows: 15 s ON/90 s OFF, repeated 6 times. A portion of sheared DNA (10%) was kept as input and the rest of the sheared DNA was immunoprecipitated with α-5-methylcytosine antibody (Diagenode), bound to magnetic beads, and was isolated. qPCR for selected genomic loci was performed and efficiency was calculated as % (me-DNA-IP/total input). Primer sequences are listed in Table S3.

### LUMA

To assess global CpG methylation, 500 ng of genomic DNA was digested with MspI+EcoRI and HpaII+EcoRI (NEB) in parallel reactions, EcoRI was included as an internal reference. CpG methylation percentage is defined as the HpaII/MspI ratio. Samples were analyzed using PyroMark Q24 Advanced pyrosequencer.

### Infinium Human Methylation 450K BeadChip analysis

All GEO datasets corresponding to Infinium Human Methylation 450 BeadChip assays were downloaded as Series Matrix files, which contain the beta value for each interrogated CpG (%mCG). We selected CpGs (probes) corresponding to L1HS to L1PA8 promoter regions by intersecting their coordinates with bedtools. A subset of younger L1PA3 elements with a 129-bp deletion in the 5′ UTR region gave rise to the L1PA2 and L1HS lineages. As this deletion is associated with altered epigenetic repression and we observed a methylation shift within the L1PA3 family in 2102Ep cells, we decided to distinguish the two L1PA3 subfamilies (L1PA3-short and L1PA3-long). The selected probes were then separated into two groups according to the family of their intersecting L1, referred to as "young L1PAs" (i.e., L1HS, L1PA2, and L1PA3-short) or "old L1PAs" (i.e., L1PA3-long, L1PA4-8). The names, coordinates and L1 annotations of the selected probes are provided in Table S5. For each sample, we compared the two groups by subtracting the median beta value of the young L1PAs from that of the old L1PAs (Δ%mCG old vs. young L1PAs), and performing a two-sided Wilcoxon rank-sum test between the two groups (young L1PAs, n = 695; old L1Pas, n = 189). Sample metadata were manually curated (Table S5).

### RNA-seq

#### RNA extraction

Total RNA was purified from the same cell pellet (split in half) as the genomic DNA for bs-ATLAS-seq by two successive cycles of TRI Reagent extraction (Molecular Research Center) and recovered in 50 μL of Milli-Q water. Subsequently, 8 μg of total RNA was treated with 2 U of TURBO DNase (Life Technologies) for 20 min at 37°C followed by a 5 min incubation step at room temperature with the DNase Inactivation Reagent. After centrifugation at 10,000 x g for 1.5 min, the supernatants containing the RNA samples were transferred to new tubes. RNA was quality-controlled and quantified by UV-spectroscopy (NanoDrop 2000), microfluidic electrophoresis (Agilent 2100 Bioanalyzer) and fluorometric Qubit RNA Assay (Life Technologies).

#### Library preparation and sequencing

Directional poly(A)+ RNA-Seq libraries were prepared using 300 ng of DNase-treated RNA using the Poly(A) mRNA Magnetic Isolation Module and NEBNext Ultra II Directional RNA Library Prep kit for Illumina (New England Biolabs) according to manufacturer's instructions. Samples were multiplexed and sequenced with 2x75 bp pair-end reads on a NextSeq 500 instrument (Illumina).

#### RNA-seq mapping

RNA-seq raw reads were trimmed to remove fragments of sequencing adapters and regions of poor sequencing quality using the sliding-window mode of Trimmomatic (v0.32)[148] and parameters recommended for paired-end reads by the Trimmomatic manual. Read quality before and after trimming was then verified using FASTQC (v0.11.2) (http://www.bioinformatics.babraham.ac.uk/projects/fastqc). Trimmed reads were mapped against the human reference genome hg38 (with GENCODE comprehensive release

29), using STAR (v2.7.5c)[152], with the following non-default parameters: –outFilterMultimapNmax 1000 (1000 alignments allowed per read-pair), –alignSJoverhangMin 8 (minimum overhang for unannotated junctions).

### Gene expression measurement

Gene expression was quantified using StringTie[172] with default settings and with -e option to quantify expression only to annotated transcripts from GENCODE annotation gtf file (comprehensive release 29).

### L1 expression measurement

Locus-level L1 expression was approximated by the level of readthrough transcription in the downstream flanking sequence, resulting from the weak L1 polyadenylation signal, as previously achieved.[9,28] Briefly, we calculated the number of unique RNA-seq reads mapped within a 1 kb-window downstream of L1 and on the same strand, and subtracted the number of unique reads mapped within a 1 kb-window upstream of L1 to eliminate signal from surrounding pervasive transcription. Then, the value was normalized by the total number of mapped reads (RPKM). More specifically, we considered only mapped R2 reads with MAPQ $\geq$ 20 since they are oriented in the same direction as the RNA fragment. They were first extracted from the bam file using samtools (samtools view -b -f 128 -F 4 -q 20). Then, the number of mapped R2 reads in a 1 kb-window upstream and downstream the L1 element and in the same orientation as L1 were counted using BEDtools (coverageBed -s). Annotated exons overlapping with these regions and on the same strand as L1 were masked. Finally, the 5′ signal was subtracted from the 3' signal to remove potential noise due to pervasive transcription, and the result was normalized by the number of mapped reads to give a value as L1 reads for 1kb per million of mapped reads (RPKM). Negative values (more 5′ signal than 3′ signal) were set to zero. As a cross-validation, the expression levels of individual L1HS copies were also measured with L1EM (v1.1) and recommended parameters.[109] To measure L1 expression aggregated at the family-level, we used TEtranscripts (v2.2.1),[164] combined with DESeq2 (v1.30.1) for differential expression analysis.[163] We further annotated the expressed elements for intactness (i.e., without stop codon in ORF1 or ORF2 according to the L1Base2 database[173]) and for published evidence of retrotransposition activity, as determined by cell culture assays or by the identification of transduction events deriving from the locus. The corresponding publications are listed in Table S2 for all L1 elements detected by bs-ATLAS-seq across all cell lines, and in Table S8 for the subset of expressed L1 elements.

### Chimeric transcript discovery

Splice junctions are counted during the mapping step and are summarized in the table SJ.out.tab from STAR. Each splice junction is characterized by its coordinates and the number of mapped reads which supports the junction. To detect chimeric transcript between an L1 and a neighboring gene, the "start" and the "end" are dissociated and separately analyze with bedtools intersect. Only splice junction for which one extremity mapped into L1 and the other into an exon (GENCODE comprehensive release 29), and supported by at least 2 uniquely mapped reads, are retained. Then uniquely and multi-mapped reads are summed and normalized by the number of mapped reads per million (RPM).

### Variant calling

Variant in epigenetic regulators were called from RNA-seq alignments using GATK tools with the HaplotypeCaller option.[158] The output VCF files were filtered with bcftools, keeping only variants supported by a minimum of 10 reads and a quality score superior to 30. To annotate and assess the functional consequences of non-synonymous mutations, we employed both VEP (Variant Effect Predictor from Ensembl) and PolyPhen-2 tools.[159,174] Variant allele frequencies were calculated from the 1KGP general population and provided by VEP. The called variants and their annotations are reported in Table S4.

### 5-Aza-2′-deoxycytidine treatment

HCT-116 cells were cultured in McCoy medium supplemented with 10% FBS, 100 U/mL penicillin and 100 µg/mL streptomycin. For 5-aza-2′-deoxycytidine (5-aza) treatment, cells were plated at a density of 100,000 cells/well in 6-well plates and treated with 5-aza at a final concentration of 1 µM for a total of 5 days. Fresh medium and drug were added daily for the first 3 days.

### ChIP-seq

#### Chromatin immunoprecipitation (ChIP)

For ChIP of the transcription factor YY1, exponentially-growing 2102Ep cells were washed twice with PBS and fixed at room temperature by addition of disuccinimidyl glutarate to a final concentration of 2 mM and incubation for 45 min, followed by two washes in PBS and addition of formaldehyde to a final concentration of 1% and incubation for 15 min. For ChIP of histone H3K4me3, 2102Ep cells were fixed by addition of formaldehyde to a final concentration of 1% directly to the cell growth medium. Fixation was stopped by addition of glycine to a final concentration of 125 mM. Fixed cells were washed once quickly and twice for 10 min each with ice-cold PBS, and collected by scraping and centrifugation for 10 min at 500 xg at 4°C. Nuclei were collected by centrifugation for 5 min at 500 xg at 4°C and resuspended at 5x10^7 cells/mL in 900 µL of ice-cold L2 buffer (50 mM Tris pH 8.0, 5 mM EDTA, 1% SDS) containing protease inhibitors. Chromatin was fragmented by sonication to an average size of 600–700 bp (typically 9 cycles of 10 s sonication, 1 min recovery on ice, using a micro-tip sonicator) and insoluble debris was pelleted by centrifugation. A 50 µL-aliquot was removed from each sample and analyzed by agarose gel electrophoresis after DNA extraction to verify fragmentation. Fragmented chromatin was diluted with 9 volumes of buffer DB (50 mM Tris pH8, 200 mM NaCl, 5 mM EDTA, 0.5% NP40), and 1 µg of anti-YY1 antibody (C15410345, Diagenode) or anti-H3K4me3 antibody (ab8580, Abcam) was added to each 1 mL of chromatin and incubated overnight at 4°C with rotation. Antibody-bound chromatin was pulled-down by addition of 25 µL of protein-A Dynabeads (Invitrogen) for anti-YY1 ChIP, or by addition of 15 µL of protein-A Sepharose for anti-H3K4me3 ChIP, incubated for 30 min at 4°C, and collected using a

magnet for dynabead-bound chromatin, or by centrifugation. Chromatin-bound beads were washed once quickly and 4 times for 5 min each with 900 μL of ice-cold buffer WB (20 mM Tris pH 8.0, 500 mM NaCl, 2 mM EDTA, 1% NP40, 0.1% SDS), followed by washing for 5 min each with 900 μL of ice-cold TE followed by 350 μL of ice-cold 10 mM Tris pH 8.0.

*Library preparation and sequencing*
For YY1 ChIP, immunoprecipitated chromatin was tagmented on beads based on the Diagenode 'TAG kit for chipmentation' protocol, using a total tagmentation time of 15 min, and sequencing libraries were prepared from tagmented samples by PCR amplification using Kapa HiFi polymerase (Roche). For H3K4me3 ChIP, immunoprecipitated chromatin was released by incubating beads three times in buffer EB (TE + 2% SDS) for 5 min at room temperature with periodic tickling, and pooling the supernatants after collection; fixation was then reverted by overnight incubation at 65°C, and DNA was directly purified using the MinElute PCR purification kit (Qiagen), eluted in 30 μL elution buffer, and sequencing libraries were prepared using the NEBNext Ultra II DNA library kit (New England Biolabs). Samples were sequenced using a paired-end strategy on a NextSeq500 instrument (Illumina).

*ChIP-seq analysis*
Sequencing reads were trimmed with cutadapt (-q 10) and were aligned to the human reference genome hg38 using bowtie2 (v2.4.1) with options –very-sensitive and –end-to-end. Peaks were called with MACS2 (v2.2.7.1) and the following parameter: -g 2.9e9 using input DNA as background. An identical procedure was applied to process our own experimental ChIP-seq data and to reanalyze publicly available datasets (ESR1 in MCF-7 and YY1 in H1 cells). For the other publicly available transcription factor datasets, peaks were computed by and obtained from the Unibind database (see paragraph below). For H3K4me3, the broad peak option (–broad) was also selected. Coverage was calculated using deeptools (bamCoverage –minMappingQuality 10 –normalizeUsing RPKM –binSize 10), and visualized in IGV.

### Transcription factor enrichment at unmethylated L1

Differential transcription factor binding between unmethylated (mCG<25%) and methylated (mCG>75%) L1HS and L1PA2 subsets was analyzed for each cell line using the Unibind enrichment command line tool (UniBind_enrich.sh, available at https://bitbucket.org/CBGR/unibind_enrichment) and the entire Unibind database (Hg38_robust_UniBind_LOLA.RDS) using the twoSets option,[95] which allows to compute differential enrichment. Unibind enrichment tool relies on the LOLA algorithm, which uses Fisher's exact test with false discovery rate correction to assess the significance of overlap in each pairwise comparison.[175] Note that for each cell line in our panel, we compared the methylated and unmethylated L1 subsets to all ChIP-seq data stored in Unibind, irrespective of the cell-type or conditions in which they were obtained. The rationale is that even if datasets from our cell line of interest are not present in Unibind, a similar cell type or condition may be represented. The region considered for TF enrichment encompasses the 300 bp upstream sequence under L1 epigenetic influence and the first 500 bp of the L1 promoter, as it shows variable methylation.

### Enrichment of genomic features

To allow a fair comparison of the associations of reactivated and non-reactivated L1HS upon 5-aza treatment with a wide range of genomic features, we used a statistical approach in which we generate a large number of controlled *in silico* randomizations of each dataset, and we express the magnitudes of each association as a *Z* score, which reflects the number of standard deviations by which the measured similarity of any pair of datasets differs from the similarity expected by chance, as previously performed.[67]

### Detection of L1-mediated epivariation

To determine whether L1 presence or its methylation status influences flanking sequence methylation, we compared the proportion of differentially methylated alleles at proximal and distal CpG sites between filled and empty alleles. We measured proximal methylation at the closest CpG in a 300 bp window upstream of L1, and distal methylation at the closest CpG 1–2 kb upstream of L1. In this approach, the empty allele is used as a proxy for the pre-insertion allele. Although we cannot completely exclude allele-specific methylation independent of L1, this possibility is mitigated by comparing proximal vs. distal methylation from the same allele, which accounts for the gradual influence of L1 on its flank. This approach also maintains the association between a given locus and the general epigenetic state of the cell line from which it was extracted. We restricted our analysis to heterozygous L1 loci with at least 5 reads covering each allele and used a two-sided Wilcoxon rank-sum test to compute the difference in methylation between the same CpG site in the empty and filled alleles, considering each read as an observation and yielding to a corresponding p value.

The first hypothesis tested was that the presence of an L1 element impacts the methylation of its proximal upstream sequence. We classified each CpG site as "influenced" if the methylation of the CpG site in question in the filled allele was statistically different from that of the empty allele (two-sided Wilcoxon rank-sum test, p < 0.05) and "not influenced" if not. Under the null hypothesis, the proportion of CpG sites labeled as "influenced" or "not influenced" is similar at proximal and distal sites. The null hypothesis is consistent with L1 presence on its own having no particular effect on the methylation status of the locus in which it is inserted. To test this hypothesis, we used a chi-squared test of independence.

The second hypothesis tested was that the methylation status of the L1 element influences the methylation of its proximal upstream sequence. Here, we categorized a CpG site as "influenced" if the methylation of the CpG site in question in the filled allele was statistically different from that in the empty allele (two-sided Wilcoxon rank-sum test, p < 0.05) <u>and</u> if the methylation level of the L1 promoter (average CpG methylation in the first 200 bp of the L1 sequence) differed by more than 30% as compared to the methylation level of the CpG site in question in the empty allele. Conversely, a CpG site was labeled as "not influenced" if the

methylation of the CpG site in question in the filled allele was not statistically different from that of the empty allele but with the methylation level of the L1 promoter still different by more than 30% as compared to the methylation level of the CpG site in question in the empty allele. Any CpG site not falling into one of the above categories was considered inconclusive, and labeled as such. Under the null hypothesis, the proportion of the 3 categories is similar at proximal and distal CpG sites. The null hypothesis is consistent with L1 methylation having no effect on the methylation state of the nearby sequence. The alternative hypothesis is that L1 epigenetic state can mediate local epivariation. To test this second hypothesis, we also used a chi-squared test of independence.

## QUANTIFICATION AND STATISTICAL ANALYSIS

Statistical analyses were performed in R and are explicitly described in each Figure legend. Unless otherwise stated:

(1) For comparing the distribution of methylation levels for groups of L1 elements that differ by a given genomic feature (e.g., in Figure 3C, the average methylation levels of individual L1HS copies in expressed genes vs. non-expressed genes), we employed the two-sided Wilcoxon rank-sum test, a non-parametric test that does not assume normality of the data, with each L1 locus considered as an independent observation. This was achieved with R function statswilcox.test.

(2) For comparing the proportion of L1s with different features between two or more independent groups (e.g., in Figure 3D, the proportion of L1 in different methylation categories and between different genomic compartments), we used a chi-squared test of independence, a non-parametric test that does not assume normality of the data, with the appropriate contingency table, and without Yates continuity correction. This was achieved with R function statschisq.test without Yate's continuity correction (correct = FALSE) as all groups were larger than 5. Chi-squared tests were reported with their chi-squared statistic value ($\chi^2$), degree of freedom (df), sample size (N), and p value (p).

More specific statistical analyses are provided in the relevant Methods sections.

