## [Document S2. Transparent peer review records for Sophie Lanciano et al · Cell Genomics]

Locus-level L1 DNA methylation profiling reveals the epigenetic and transcriptional interplay between L1s and their integration sites

Sophie Lanciano, Claude Philippe, Arpita Sarkar, David Pratella, Cécilia Domrane, Aurélien J. Doucet, Dominic van Essen, Simona Sacconi, Laure Ferry, Pierre-Antoine Defossez and Gael Cristofari

Summary

Initial submission: Received : June 02, 2023

Scientific editor: Sara Rohban

First round of review: Number of reviewers: 3
Revision invited : July 20, 2023
Revision received : November 21, 2023

Second round of review: Number of reviewers: 3
Accepted : January 09, 2024

Data freely available: YES

Code freely available: YES

This transparent peer review record is not systematically proofread, type-set, or edited. Special characters, formatting, and equations may fail to render properly. Standard procedural text within the editor's letters has been deleted for the sake of brevity, but all official correspondence specific to the manuscript has been preserved.

Referees' reports, first round of review

Reviewer #1: In this manuscript, Lanciano et al 2023 apply a suppression PCR-based method for locus-specific L1 DNA methylation analysis, termed, bs-ATLAS, to a panel of human cell lines and resolve individual L1 DNA methylation profiles genome-wide. Some results with the short read-based bs-ATLAS are verified using Oxford Nanopore Technologies (ONT) long-read DNA sequencing and methylation profiling. The work is interesting to the L1 field, although somewhat limited in scope by the exclusive use of cultured cell lines rather than human tissues. Indeed, results obtained using cell lines are over-interpreted without fully considering the developmental context of DNA methylation establishment and maintenance in vivo. The finding that L1 insertions can impact the flanking epigenetic landscape is quite interesting, but the general interest of the paper would be expanded by experimental evidence that the presence of an L1 can functionally impact the regulation of nearby genes. Over all, the work is high quality and describes a novel method that appears quite robust, but which may be less attractive than long read DNA methylation analysis.

Major points:

1. The analyses in this paper were undertaken on a panel of 12 human cell lines, including normal primary fibroblasts (two fetal, one foreskin), artificially immortalized and transformed cells, cancer cell lines, and human embryonic stem cells and embryonal carcinoma cells. Analysis of human tissues is conspicuously absent. It would greatly broaden the interest and deepen the biological relevance of the paper to include a couple of tumor/non-tumor patient samples, or a normal tissue reported to accommodate elevated L1 activity (eg, brain) vs a control tissue.
2. Throughout the paper, the developmental dynamics of DNA methylation and its effectors, the DNA methyltransferases, are overlooked. The most striking example is on Page 7, lines 9-12, where Dnmt3b activity is evoked to explain the methylation of genic L1 elements in K-562 cells. However, Dnmt3b is a de novo DNA methyltransferase which establishes DNA methylation marks during embryonic development that subsequently are inherited mitotically upon cell division and maintained in somatic cells by the maintenance DNA methyltransferase DNMT1. Indeed, the referenced publications on Dnmt3b recruitment following transcription coupled H3K36me3 deposition all appear to describe its activity in embryos or embryonic cell types. The authors should provide experimental evidence that Dnmt3b is expressed in K-562 cells and that loss of Dnmt3b results in genic L1 demethylation, or soften their conclusion.
3. Similar to the above point, the section discussing the relationship between YY1 binding and DNA methylation does not consider the temporal dynamics of DNA methylation during mammalian development. The embryonal cell types are used to represent the pluripotent state, before global establishment of somatic methylation patterns in the gastrulating embryo. It is entirely feasible that YY1 binds unmethylated L1 loci in pluripotent cells and enables accurate L1 transcription initiation, but during differentiation YY1 either directly or indirectly recruits the de novo DNA methylation machinery, facilitating establishment of L1 methylation genome-wide in

the somatic lineages. The results obtained in the present study therefore are not irreconcilable with the model presented in reference 63 (Sanchez-Luque et al), that YY1 binding to the L1 promoter mediates its de novo DNA methylation during development. The discussion of this result should be softened, as it currently implies that this model has been overturned. Of course, the YY1 binding site itself, rather than binding of the YY1 transcription factor, could instead be required for establishment of DNA methylation as acknowledged both here and in reference 63.

4. In any cases, were the CpGs impacted by local L1-induced epivariation within regulatory elements of other genes or TEs? Experimental evidence for functional impact of L1-mediated epivariation would greatly elevate the paper and provide greater interest for a general audience.

5. Were replicate bs-ATLAS assays performed for each cell line? For the figure panels where statistics are performed, it is unclear but it appears that p-values were determined by considering each individual L1 locus as a biological replicate. Can the authors elaborate on their choice of statistical methods and how the statistics should be interpreted?

Minor Points:

1. Some context about the biology of the different cell lines would be helpful. There must be datasets on their overall DNA methylation and chromatin landscape that could help put the L1 results into context. Indeed, the paper is over-all highly L1 focused, and fails to incorporate existing data about global DNA methylation and the general cellular environment of the cells under study. This is particularly relevant to the K-562 cells, which are remarkably hypomethylated. Are there mutations in DNA methyltransferase genes or their interacting factors known to underpin this phenotype?

2. The PCR validation of previously undetected insertions is an excellent control. However this is not mirrored by any attempt to confirm that reference insertions not detected in each cell line are truly absent. The authors state that ~90% of reference insertions were detected and the remaining 10% were likely absent, but they could actually obtain an exact statistic for the sensitivity of their method by determining exactly which reference insertions are present and absent. Surely there must be WGS data available for at least some of these commonly used cell lines that could be analysed for L1 content?

3. Figure S11: the correlation between Atlas-bis-seq and ONT is strong, but there are some conspicuous outliers. Did the authors examine these loci and associated data in more detail to determine the cause of discordant results?

4. Methylation levels across the L1 sequence described on page 8, lines 11-14, were previously described in human patient samples by Ewing et al. 2020 (ref 110). That publication should be acknowledged here.

Reviewer #2: Lanciano and colleagues explore the relationship between L1 promoter methylation and transcription factor (TF) binding in a substantial panel of cultured human cells. It is very interesting how these parameters, alongside genomic context, combine to regulate L1 transcriptional activity. The standard of the presented analyses is exceptionally high. I commend the authors, and particularly whoever assembled the many figure panels showing L1 analyses, for the quality of their work. One caveat, emphasised by the variation in L1 regulatory "rules" amongst cultured cells, is that some of the findings may not carry over to other cell types or to in vivo biology, but there L1 regulation is also much more difficult to study. Overall it is a thoughtful and important manuscript that tells us much about L1 regulation and was a pleasure to consider.

Main comment:

The analysis of YY1 binding and L1 methylation in Fig. 5 does a nice job of showing that the YY1 motif is associated with methylation, in agreement with PMID: 31230816. This agreement could be stated more clearly, even if YY1 binding to that motif is associated with hypomethylation. It is very difficult to untangle that second relationship, because YY1 is both a repressor and an activator (Yin Yang 1), depending on context, and will bind unmethylated DNA more effectively than methylated DNA. TF binding is also dynamic. It remains possible that YY1 binding influences methylation and then, once methylated, no longer binds to a given locus, and this could be noted as well if the authors agree it a reasonable point.

I would not propose additional experiments, but to strengthen this section the authors could:

- Present the analysis for Fig. 5E in the same style as Fig. S5C, splitting the YY1 binding +/- into YY1 motif presence +/-, showing whether the YY1 motif is associated with methylation in other cell types.
- Present the analysis for Fig. 5E, except splitting the data instead into intragenic / intergenic elements. The reason for this is that almost all of the elements studied in PMID: 31230816 were deliberately chosen for their position in intergenic regions, away from the influence of genic methylation (which the authors have shown here can make a big difference to L1 methylation). It seems the "rules" are different in YY1 and 2102Ep, and that may be borne out by doing this intragenic / intergenic analysis.
- The H1 and 2102Ep methylomes are very different. In H1, it does not seem that YY1 binds L1 extensively (Fig. 5B) and there isn't H3K4me3 data available to test whether this binding results in higher transcription or not. The opposite is true in 2102Ep, where the data are much more clear. In HEK293 cells (Fig S5B) the YY1 bound young L1s are significantly MORE methylated than the unbound L1s. I think it would be fair to say that the rules around YY1 binding and L1 methylation are not easy to generally define without perturbing the system. page 9, line 44, refers to "embryonic stem cells or embryonal carcinoma cells" yet these relationships are only really clear for 2102Ep. Perhaps then the authors could amend this text to "embryonal

carcinoma cells", where the data are clearest, and make a note of future work being required to dissect the YY1 / L1 methylation relationship.

(note: my lab published a previous study evaluating the relationship between the L1HS YY1 motif and methylation PMID: 31230816 and a reader should know that when considering whether these points are reasonable)

Other Critiques / questions, in no particular order.

- 1) Line 6, page 6 (authors' page numbering): "the youngest L1HS family appears hypermethylated" could say this is in agreement with a prior analysis of human tissues (PMID: 33186547, Figure S2A)
- 2) Line 24, page 9: please state the absolute median % mC differences for H1 and 2102Ep between YY1-bound and YY1-unbound elements.
- 3) Fig. 7. The authors should either perform H3K4me3 ChIP-seq on these HCT-116 cells to measure L1 activation more comprehensively OR note a caveat that the analysis may have overlooked transcriptionally induced L1s that do not generate 3' read-through transcription.
- 4) Fig. 6B. The numbers of L1HS copies displayed suggests these are not full-length. If so, how are their methylation levels assessed?
- 5) Line 23, page 7: the authors may note that sloping shores extending into genomic flanks have previously been identified at specific L1 examples (e.g. PMID: 31230816, Fig. 3B), even though I agree that this has not been done systematically to my knowledge.
- 6) The Methods state how the YY1 peaks were defined for 2102Ep, but is this bioinformatic approach also applied to the YY1 ChIP-seq data from ENCODE? How clear are these YY1 peaks in H1? It seems like there is relatively little enrichment for YY1 binding to L1HS in H1 cells in general (Fig. 5B) and, although there are beautiful peaks shown for YY1 in the 2102Ep data (e.g. in Fig. S5E) there aren't any examples shown like this for H1. Could one be provided in supplement?
- 7) Line 19, page 7: this is also consistent with PMID: 31230816.
- 8) Fig. 6D, top right. Why is this element considered to be unexpressed? It very much looks like it has a H3K4me3 peak.

Geoff Faulkner (University of Queensland)

Reviewer #3: This important study from Lanciano, Philippe and others is an important step towards understanding the role that full-length LINE-1 elements play in altering the epigenetic and transcriptional landscape of cells, the variation in their effects and expression at a given locus, and some of the mechanisms behind these changes. The authors nicely detail an important technique, bs-ATLAS, for deciphering the methylation status of a discrete LINE-1 locus, and use this and multiple orthogonal techniques to characterize 12 human cell lines ranging from embryonic stem cells to "normal" cell lines to cancer cell lines. The data are striking and show obvious patterns of methylation change related to the CpG island in the 5'UTR of the elements, depict the effects of these methylation differences on the proximal genomic locus of the insertion, and importantly decouple methylation from expression, with the importance of the epigenetics at the locus playing a role in L1 expression. The experiments are well thought out, the findings are well explained and not overstated, and the data are important to the field.

Major Comments:

1) Overall, the explanation of statistical tests and rationale for their use could use clarification. Specifically, there should be an extensive statistics section within the STAR methods section. Although the statistics are somewhat contained within figure legends, there is not adequate room for discussion of methods, and much of the paper's findings depend on association between L1 methylation and the surrounding locus. This seeming lack of testing for certain purported enrichments within the data set should be corrected. Examples:

a) In section "L1 DNA methylation can be influenced by genic activity" on page 6, line 35: the authors state "In K-562 cells, methylated L1HS elements are enriched in genes". The authors should report statistical values for enrichment and test used in the text or in the figure panel (Figure 3A); moreover the statistics should account for the number of events in intragenic vs. intergenic space, and perhaps use permutations to test the rigor of their association.

b) Although the authors depict an association between the proximal epivariation between L1 alleles in Figure 4, there is no direct test of the effect of the TE itself on the surrounding locus, and the association observed does not have a statistical framework to rely upon. Moreover, on page 8, line 15, the authors state that "the influence of an L1 element on the flanking region could only be assessed if the methylation state of the L1 promoter differs from that of its cognate empty allele". This would appear to bias the results to only examine the 37/78 that show a difference in between the L1 insertion and the empty site. Perhaps the retrotransposition itself affected the methylation status upstream. Perhaps the other 41 loci are indicative of this being a pseudo random event. Statistics and a clearly defined question are important to draw more generalizable results. Again, controlling for the background genomic noise is important, so permutations could be helpful.

Minor comments:

2) The embryonal carcinoma 2102E cell line has a distinct L1 hypomethylation pattern when compared to K-562; although the pattern is likely due to the poised/early developmental state, this finding lacks a clear explanation in the paper. A clearer breakdown of why H1 and 2102E are

different than the other lines would be helpful for a broader audience.

3) For active elements (Figure 6C) what publications are you using to define this activity? This was not apparent in the manuscript, nor in the STAR methods.

Authors' response to the first round of review

Reviewer #1:

In this manuscript, Lanciano et al 2023 apply a suppression PCR-based method for locus-specific L1 DNA methylation analysis, termed, bs-ATLAS, to a panel of human cell lines and resolve individual L1 DNA methylation profiles genome-wide. Some results with the short read-based bs-ATLAS are verified using Oxford Nanopore Technologies (ONT) long-read DNA sequencing and methylation profiling. The work is interesting to the L1 field, although somewhat limited in scope by the exclusive use of cultured cell lines rather than human tissues. Indeed, results obtained using cell lines are over-interpreted without fully considering the developmental context of DNA methylation establishment and maintenance in vivo. The finding that L1 insertions can impact the flanking epigenetic landscape is quite interesting, but the general interest of the paper would be expanded by experimental evidence that the presence of an L1 can functionally impact the regulation of nearby genes. Overall, the work is high quality and describes a novel method that appears quite robust, but which may be less attractive than long read DNA methylation analysis.

We thank the Reviewer for their time, for their thoughtful comments and for recognizing the high quality of our study. We value their perspective on the limitations of using cultured cell lines, and have conducted new analyses to address this concern. We have validated and extended our observations to a large set of human tissues and conditions, as detailed in our response to comment #1.1 below. We hope that these additional analyses will further strengthen our manuscript.

We fully acknowledge the importance of developmental dynamics in the regulation of DNA methylation. While our study focuses on DNA methylation patterns in cell lines, we recognize that these patterns may differ from those observed in vivo due to the developmental context. We addressed this limitation in the original 'Limitations' section, though it may not have been adequately emphasized. We have now revised this section to make it clearer (p. 16):

However, in other cases this remains challenging, due to the dynamic nature of DNA methylation throughout development and differentiation, which is difficult to capture using cell lines, even though we could associate changes in L1 family-specific methylation profiles with changes in pluripotency (Figure 2). Indeed, observed methylation profiles could have been established in a different cell type through factors that are no longer present and could have been maintained ever since.

We also acknowledge the unique advantages of long-read sequencing for DNA methylation analysis. However, we believe that different technologies, including short-read sequencing, long-read sequencing, and methylation arrays, each offer distinct advantages and limitations. For this reason, we combined these three orthogonal technologies in our study, and found this strategy to be fruitful. This point is addressed in a revised paragraph of the Discussion section (p. 14):

*Bs-ATLAS-seq offers specific advantages, including excellent cost-effectiveness, since it requires only 10-20 million reads per sample, and versatility with regard to genomic DNA quality, since it works with partially fragmented genomic DNA, as is typically the case in clinical samples. Bs-ATLAS-seq can be combined with other emerging approaches, such as nanopore long-read sequencing, to haplotype-resolve DNA methylation over entire loci^{47,92,172} (as illustrated by our allele-specific methylation analysis, **Figure 4**). It can also be used in conjunction with more established technologies, such as methylation arrays, which remain a platform of choice for large clinical studies of the epigenome (as illustrated in our screen for differential methylation between L1 families, **Figure 3**).*

Major comments:

1.1. The analyses in this paper were undertaken on a panel of 12 human cell lines, including normal primary fibroblasts (two fetal, one foreskin), artificially immortalized and transformed cells, cancer cell lines, and human embryonic stem cells and embryonal carcinoma cells. Analysis of human tissues is conspicuously absent. It would greatly broaden the interest and deepen the biological relevance of the paper to include a couple of tumor/non-tumor patient samples, or a normal tissue reported to accommodate elevated L1 activity (eg, brain) vs a control tissue.

Before addressing this comment, we would like to emphasize a critical aspect of our study. A key element of our work was the ability to examine the association of L1 DNA methylation data with a vast array of genomic and epigenomic features, providing new mechanistic insights into the interplay between L1 DNA methylation and the host genome. This comprehensive analysis was only possible due to the availability of publicly available datasets in matched samples. Access to these datasets enabled us to investigate a multitude of features, which would not have been feasible in a single study, particularly with human samples, and to perform perturbation experiments. While we agree with the Reviewer's general concerns about the limitations of cell lines, cellular experimental models remain invaluable tools for exploring human L1 regulation, as also emphasized by Reviewer #2.

To effectively demonstrate the potency of cell lines to draw biologically relevant conclusions, we validated and extended our initial observation that some cell types of embryonal origin can show a dramatic difference of methylation between the youngest primate-specific L1 families, and older ones, and concluded that such a difference is highly specific for pluripotent stem cells, particularly in a naïve state, and to trophoblasts and placenta, but is not observed in most human tumors.

New results (p. 7):

A striking observation is that most cell lines, including those of cancer origin, show higher levels of DNA methylation for young L1PAs than for old L1PAs, while cells of embryonic origin (2102Ep and, to a lesser extent, H1) have the opposite profile, with young L1PAs being less

methylated than old L1PAs (**Figure 2A** and **Figure S2G**, top). To extend these observations to a broader spectrum of human conditions, cell types, or tissues, we sought to exploit publicly available datasets generated using Illumina Infinium Human Methylation 450 (450K) BeadChip arrays, which are widely used for large clinical studies. For this purpose, we identified probes that overlap with either young ($n=695$) or old ($n=189$) L1PA elements (see the 'Methods' section for details of how the two categories were defined, and **Table S5**). Despite the limited number of interrogated CpG dinucleotides and some degree of likely cross-hybridization among closely related L1 loci, the assay effectively captures the lower methylation of young L1PAs as compared to old L1PAs in 2102Ep (**Figure S2G**, bottom). Similarly, it reliably detects the elevated methylation of young L1PAs compared to old L1PAs in normal fibroblasts (BJ, IMR90, MRC-5) as well as in some cancer cell lines (MCF-7, HeLa-S3, HepG2) (**Figure S2G**, bottom). For other cell types, the overall trend was also consistent with *bs-ATLAS-seq* results but the difference between the two L1 groups was not statistically significant.

New Figure S2G:

(G) Comparison of methylation levels for young (L1HS to L1PA3-short) and older (L1PA3-long to L1PA8) elements across the different cell lines using *bs-ATLAS-seq* or BeadChip arrays. Top: *bs-ATLAS-seq* (each data point is an individual L1 element), bottom: 450K Illumina BeadChip arrays (each data point is an individual L1 probe). Publicly available 450K Illumina BeadChip array datasets are listed in **Table S5**.

New results (p. 7):

Encouraged by this result, we systematically measured differences of methylation between young and old elements in 450K methylation array data publicly available in GEO (~12,000 samples, **Table S5**). Remarkably, consistent with *bs-ATLAS-seq* results, and in agreement with prior nanopore analysis of a couple of human tissues⁴⁷, we observed that the methylation of young L1PAs is high, and generally similar to or higher than that of old L1PAs in most situations. In contrast, young L1PAs appear hypomethylated in pluripotent stem cells (ESC, iPSC), trophoblast, embryonal carcinoma cells, seminoma, placenta, fetal membranes and hydatidiform moles, situations related to early embryogenesis, extra-embryonic tissues or male germ line tumors (**Figure 2B** and **Figure 2C**). The methylation difference between young and old L1PAs increases when primary cells are reprogrammed into iPSC (**Figure 2D** and **Figure**

2E, left) or when ESC or iPSC are treated with a LIF+3i cocktail that reverts conventional primed stem cells to a naive pluripotent state (Figure 2D and Figure 2E, right). In contrast, the methylation difference between the two groups is reduced when ESC and iPSC are differentiated (Figure 2D and Figure 2E, middle).

Collectively, these data demonstrate that L1 DNA methylation is cell type- and family-specific, with young L1PA hypomethylation being rare at the family level and occurring mainly in pluripotent cells, early development, and extra-embryonic tissues. In contrast, young L1PA hypermethylation is the most common situation, even in tumors, supporting the notion that L1 activity in tumors or normal somatic tissues results from a very limited number of L1 progenitors that escape epigenetic repression^{107,116,119,121,130,186}.

New Figure 2:

Figure 2 – L1 promoter DNA methylation is cell-type- and family-specific.

(A) Distribution of DNA methylation levels (% mCG + hmCG) obtained by bs-ATLAS-seq, by family and cell type. Boxplots represent the median and interquartile range (IQR) $\pm 1.5 \cdot IQR$ (whiskers). Outliers beyond the end of the whiskers are plotted individually.

(B) Differential DNA methylation between young (L1HS to L1PA3-short) and older (L1PA3-long to L1PA8) L1PA elements in all Illumina 450K array datasets publicly available in GEO (~12,000 samples, Table S5). The x-axis of the plot represents the methylation level of the

young L1PA elements, and the y-axis the difference of methylation between young and older L1PA elements (median difference of β -values). Samples with a p-value < 0.05 (two-sided two-sample Wilcoxon test) are colored in pink, and in grey otherwise. Samples with the top-20 p-values among cells or tissues are labeled with their names. Note that each data point represents the aggregated values (median) of all related samples and the size of the data point represents the number of samples aggregated.

(C) Differential DNA methylation between young and older L1PA elements (see above for their definition). The x-axis of the volcano plot represents the difference of methylation between the two groups of L1PA elements (median difference of β -values), and the y-axis the $-\log_{10}(p\text{-value})$ (two-sided two-sample Wilcoxon test). Cancerous and noncancerous samples are colored in yellow and blue, respectively, and those with the top-10 p-values among cells or tissues are labeled with their names. Note that each data point represents the aggregated values (median) of all related samples and the size of the data point represents the number of samples aggregated.

(D, E) Differential methylation between young and old L1PA elements is associated with pluripotency. Left, reprogramming of primary cells into iPSC (left panel, data from GSE51921 and GSE76372); middle, differentiation of ESC and iPSC into differentiated cells (data from GSE31848 and GSE116754); right, treatment of ESC or iPSC with LIF and 3i, a cocktail that promotes a naive state rather than a primed state (data from GSE65214). (D) Extracted data points from the volcano plot shown in (C). (E) Boxplots represent the median and interquartile range (IQR) $\pm 1.5 * IQR$ (whiskers) with overlaid individual data points indicating the median difference of methylation between young and old elements for individual samples. P-values are from two-sided two-sample Wilcoxon tests.

See also Figure S2, Table S3, and Table S5.

1.2. Throughout the paper, the developmental dynamics of DNA methylation and its effectors, the DNA methyltransferases, are overlooked. The most striking example is on Page 7, lines 9-12, where Dnmt3b activity is evoked to explain the methylation of genic L1 elements in K-562 cells. However, Dnmt3b is a *de novo* DNA methyltransferase which establishes DNA methylation marks during embryonic development that subsequently are inherited mitotically upon cell division and maintained in somatic cells by the maintenance DNA methyltransferase DNMT1. Indeed, the referenced publications on Dnmt3b recruitment following transcription coupled H3K36me3 deposition all appear to describe its activity in embryos or embryonic cell types. The authors should provide experimental evidence that Dnmt3b is expressed in K-562 cells and that loss of Dnmt3b results in genic L1 demethylation, or soften their conclusion.

We showed in K-562 cells, and in the 8 other cell types of our panel that gene body transcription and methylation of intragenic L1 elements are strongly associated (Figure 3). We concur with the reviewer regarding the importance of the developmental dynamics of DNA methylation and its effectors to fully comprehend these observed methylation patterns. As suggested by the Reviewer, we checked the expression and mutational status of DNMTs across the cell lines of the panel to substantiate the possibility that the model previously proposed in embryonic cells could also work in other cell types. We observed that both *de novo* and maintenance DNA methyltransferases are expressed at the RNA level in K-562 cells and other cell types (Reviewer Figure 1, and Figure S2F), consistent with previous protein-level observations in K-562 cells (Fig. 1D in Garzon R. et al. Blood

2009, PMID: 19211935). In light of these findings, we have made revisions to the text as follows (p. 8):

Mammalian CpG methylation is directed to gene bodies through the deposition of H3K36me3 during transcription and the subsequent recruitment of the de novo DNA methyltransferase (DNMT) Dnmt3b, at least in mouse early development and germ cells^{12,86,100,102}. Consistent with this, methylated L1s in K-562 cells are mostly found in transcribed regions with high H3K36me3 and DNA methylation (Figure 3F, Figure 3G and Figure S3C). Moreover, the de novo DNMTs, DNMT3a and DNMT3b, are expressed in all the cell lines of the panel (Figure S2F). This suggests that a similar mechanism may operate in most cell types, although the DNMTs involved need to be further investigated.

Reviewer Figure 1:

Reviewer Figure 1 - Expression profile of DNMTs in K-562 cells (polyA+ RNA-seq). Note that the major isoform in K-562 is DNMT3A2.

New Figure S2F:

(F) Heatmap of epigenetic effector expression across the different cell lines. TPM, transcripts per million of reads. Superimposed circles indicate homozygous (full circle) or heterozygous (half circle) variants detected by the VEP tool in RNA-seq data. Note that rare (AF < 0.01 in the global 1KGP population) and predicted as probably damaging (using the PolyPhen-2 tool) variants are represented, as well as variants with unknown AF and PolyPhen prediction. See Table S4 for the full list of called variants.

1.3. Similar to the above point, the section discussing the relationship between YY1 binding and DNA methylation does not consider the temporal dynamics of DNA methylation during mammalian development. The embryonal cell types are used to represent the pluripotent state, before global establishment of somatic methylation patterns in the gastrulating embryo. It is entirely feasible that YY1 binds unmethylated L1 loci in pluripotent cells and enables accurate L1 transcription initiation, but during differentiation YY1 either directly or indirectly recruits the de novo DNA methylation machinery, facilitating establishment of L1 methylation genome-wide in the somatic lineages. The results obtained in the present study therefore are not irreconcilable with the model presented in reference 63 (Sanchez-Luque et al), that YY1 binding to the L1 promoter mediates its de novo DNA methylation during development. The discussion of this result should be softened, as it currently implies that this model has been overturned. Of course, the YY1 binding site itself, rather than binding of the YY1 transcription factor, could instead be required for establishment of DNA methylation as acknowledged both here and in reference 63.

We apologize if our previous wording implied that we were overturning the recent model proposed by Sanchez-Luque *et al.* This was not our intention, and we appreciate the opportunity to clarify our findings. In line with Reviewer #2's comment #2.1, we have further analyzed our L1 methylation data and found that among unbound L1 elements, those lacking the YY1 binding site are indeed often less methylated than those containing the site (updated **Figure S5C**). We thus completely agree on the scenario proposed by the Reviewer. Further research is needed to elucidate the factors that may trigger YY1 release from methylated L1, especially considering our analysis of YY1 ChIP-seq experiments in H1 cells, which suggests that L1 methylation alone is not sufficient for YY1 release (new **Figure S5D**). Given this uncertainty, we have refrained from speculating on this aspect of the model in the revised text (p. 11):

*We confirmed that the same locus (chr13Δ31_{L1}), as well as other L1 elements carrying a similar deletion analyzed in the same study, are hypomethylated in H1 and 2102Ep embryonic stem cells and also in HepG2 cells (Figure 5E, blue labeled data points). Therefore, we analyzed separately L1 elements lacking the YY1 motif, those possessing the motif but being unbound by YY1, and those with the motif and being bound (Figure S5C). Interestingly, among unbound L1 elements, those lacking the YY1 motif are less methylated than those with the YY1 motif, in H1, HepG2 or HEK-293 cells (Δ%*mCG*=21%, Δ%*mCG*=17% and, Δ%*mCG*=11%, respectively). In 2102Ep cells, YY1-bound L1 elements exhibit the active chromatin mark H3K4me3 and higher expression levels compared to unbound elements (Figure 5D, Figure S5E and Figure S5F). In contrast to 2102Ep embryonal carcinoma cells, the majority of YY1-bound L1HS and L1PA2 are hypermethylated in the embryonic stem cells H1, and only a minor fraction is hypomethylated and marked by H3K4me3 (Figure S5D and Figure S5F). Considering that 2102Ep cells are nullipotent and blocked in an undifferentiated embryo-like state⁷³, and that H1 cells under our growth conditions are likely in a primed state (see above), our observations are thus consistent with a model whereby YY1 preferentially binds unmethylated L1 loci in naïve pluripotent cells, thereby enabling accurate L1 transcription initiation, but subsequently mediates L1 de novo DNA methylation upon cellular differentiation during development¹¹⁹, eventually associated with a loss of YY1 binding. Alternatively, alterations of the YY1 motif*

may incidentally affect other pathways targeting the same sequence and leading to L1 hypomethylation ¹¹⁹.

Updated Figure S5C and new Figure 5D:

(C) DNA methylation levels of L1HS elements with (+) or without (-) YY1 binding motifs in their 5' UTR, and actually bound (+) or not (-) by YY1 in embryonal cell lines (H1 and 2102Ep) and other cell lines for which matched YY1 ChIPseq were also publicly available (K-562, HCT116, HepG2, HEK-293). The number of L1HS copies in each subset (n) is indicated at the bottom of the plot. The data are identical to those presented in Figure 5E, but the loci unbound by YY1 were divided into two categories based on the presence or absence of the YY1 motif.

(D) Heatmap displaying L1 methylation (bs-ATLAS-seq), as well as YY1 and H3K4me3 binding (ChIP-seq), at the 5' junction (-1 to +0.5 kb) of L1HS and L1PA2 elements in H1 cells. Loci are sorted by increasing levels of L1 methylation. ChIP-seq signal represents the number of normalized reads per 10-bp bin.

1.4. In any cases, were the CpGs impacted by local L1-induced epivariation within regulatory elements of other genes or TEs? Experimental evidence for functional impact of L1-mediated epivariation would greatly elevate the paper and provide greater interest for a general audience.

The concept of TE methylation spreading has been around since the mid-1980s, with initial observations in endogenous retroviruses of mice (Jähner and Jaenisch, Nature, 1985). This phenomenon has been extensively studied in plants, where it can extend for several kilobases. However, our work represents the first comprehensive investigation of L1-mediated epivariation in humans at the genome-wide scale, revealing that it is indeed a frequent but relatively short-range event. By quantifying these events, our study challenges the common assumption that DNA methylation of young L1s has a widespread impact

on nearby DNA methylation. We believe that this result, in itself, holds significant interest for a general audience.

Regarding the potential impact of L1-induced epivariation on regulatory elements, very few ENCODE CREs are located in the vicinity of the L1s in questions, hindering statistical testing. However, in our search for transcription factors differentially bound to hypomethylated versus hypermethylated loci, we intentionally considered a window ranging from -300 bp to +500 bp relative to the L1 5' end, encompassing the region potentially influenced by the L1 (**Figure 5A**). This approach led to the identification of ESR1, which binds to both upstream and internal regions of many L1 elements. While we cannot conclusively distinguish between the influence of ESR1 internal binding sites and upstream ones, our findings demonstrate that ESR1 binding is associated with L1 hypomethylation and drives the transcription of L1 chimeric transcripts (**Figure 5**).

1.5. Were replicate bs-ATLAS assays performed for each cell line? For the figure panels where statistics are performed, it is unclear but it appears that p-values were determined by considering each individual L1 locus as a biological replicate. Can the authors elaborate on their choice of statistical methods and how the statistics should be interpreted?

Considering the high level of concordance between replicates obtained in MCF-7 cells during the development of the method (**Figure S1D-E**), we performed a single bs-ATLAS-seq assay for each cell line. The robustness of L1 methylation profiles obtained by bs-ATLAS-seq was further validated by nanopore sequencing (**Figure S1K**).

The Reviewer is absolutely correct that in many figure panels, we compare the distribution of methylation levels for groups of L1 loci that differ by a given genomic feature. For instance, in **Figure 3C**, we compared the average methylation levels of individual L1HS copies in expressed genes vs nonexpressed genes. We employed the Wilcoxon test for this comparison, a non-parametric test that does not assume normality of the data. We have now clarified the statistical tests used for each panel. For instance, in the legend of Figure 3:

*In panels (B) and (C), boxplots represent the median and interquartile range (IQR) $\pm 1.5 * IQR$ (whiskers). Outliers beyond the end of the whiskers are plotted individually. * $p < 0.05$, ** $p < 0.01$, *** $p < 0.001$, and **** $p < 0.0001$, two-sided two-sample Wilcoxon test (green for increase and black for decrease), with each L1 locus being considered as an observation.*

Additionally, we have included a dedicated section in the Methods section specifically addressing statistical methods, as suggested by Reviewer #3.

Minor Points:

1.6. Some context about the biology of the different cell lines would be helpful. There must be datasets on their overall DNA methylation and chromatin landscape that could help put the L1 results into context. Indeed, the paper is over-all highly L1 focused, and fails to incorporate existing data about global DNA methylation and the general cellular environment of the cells under study. This is particularly relevant to the K-562 cells, which are remarkably hypomethylated. Are there mutations in DNA methyltransferase genes or their interacting factors known to underpin this phenotype?

We appreciate the reviewer's suggestion to provide more context about the epigenetic state of the different cell lines under study. We had previously measured global DNA methylation levels using LUMA for a subset of cell lines, including the two with exceptionally low L1 methylation (p. 6):

In two cell lines, the embryonal carcinoma cells 2102Ep and the chronic myeloid leukemia cells K-562, L1HS elements are exceptionally hypomethylated but with very distinct epigenetic contexts. In 2102Ep cells, hypomethylation is restricted to the young L1 families (L1HS, L1PA2, and to some extent L1PA3), while older L1 elements and the rest of the genome show high levels of methylation (Figure 2A and Figure S2E). In contrast, K-562 cells display a global hypomethylation phenotype that affects all L1 families and reflects genome-wide hypomethylation, down to levels as low as those observed in HCT-116 cells with DNMT1 and DNMT3B inactivating mutations (Figure 2A and Figure S2E).

As suggested by the Reviewer, we also investigated the general cellular environment of the cells under investigation by examining the expression levels and mutational status of a panel of key epigenetic regulators (p. 6, see new Figure S2F in answer #1.2).

1.7. The PCR validation of previously undetected insertions is an excellent control. However, this is not mirrored by any attempt to confirm that reference insertions not detected in each cell line are truly absent. The authors state that ~90% of reference insertions were detected and the remaining 10% were likely absent, but they could actually obtain an exact statistic for the sensitivity of their method by determining exactly which reference insertions are present and absent. Surely there must be WGS data available for at least some of these commonly used cell lines that could be analysed for L1 content?

We appreciate the reviewer's valuable suggestion, which has enabled us to refine our assessment of bs-ATLAS-seq sensitivity. Utilizing publicly available whole-genome sequencing data for four cell lines, we employed the MELT deletion tool to determine whether reference L1HS elements not detected by bs-ATLAS-seq were genuinely absent from the sample or missed by the method (new Figure S1H). In a complementary approach, we identified L1HS-PreTa elements fixed in the human population, according to the 1000 Genomes Project, and used the percentage of recovery of these elements as a proxy for bs-ATLAS-seq sensitivity (new Figure S1G). Both strategies yielded similar estimates, confirming the high sensitivity of the method (p. 5):

Undetected reference L1HS elements may be absent from the assayed samples, or may be present but missed by the assay. By calculating the recovery rate of L1HS-PreTa elements, a lineage considered to be fixed in the human population¹²⁵, we estimated sensitivity at $97.2\% \pm 1.7\%$ (Figure S1G). As a complementary approach, we verified the presence or absence of the undetected reference L1HS copies in whole genome sequencing data of four cell lines (K-562, HCT-116, HepG2 and MCF-7), and obtained a sensitivity at $97.6\% \pm 2.1\%$ (Figure S1H).

New Figures S1G and S1H:

(G) Evaluation of *bs-ATLAS-seq* sensitivity using reference L1HS-PreTa elements defined as fixed in the human population by the 1000 Genomes Project (1KGP). Sensitivity: $97.2 \pm 1.7\%$ (mean \pm SD).

(H) Validation of true negative L1HS elements and determination of *bs-ATLAS-seq* sensitivity using publicly available whole genome sequencing (WGS) data from four cell lines. Sensitivity: $97.6 \pm 2.1\%$ (mean \pm SD).

1.8. Figure S1I: the correlation between Atlas-bis-seq and ONT is strong, but there are some conspicuous outliers. Did the authors examine these loci and associated data in more detail to determine the cause of discordant results?

We thoroughly inspected the read alignments for the five outliers but found no apparent artifacts. Three of these loci (in the lower right quadrant of the plots shown in Figure S1I) originated from HCT116 cells. For the 5-aza treatment experiments shown in Figure 7, we assayed two independent biological replicates of DMSO-treated control HCT-116 cells using *bs-ATLAS-seq*. This allowed us to investigate whether the discrepancy could result from technical variations in *bs-ATLAS-seq* or from inherent variability in the methylation of these specific loci. However, we observed a high degree of methylation concordance between the three *bs-ATLAS-seq* values (**Reviewer Table 1**), suggesting that neither biological nor technical variation is the cause of the discordance.

name	bs-panel	bs-dmso-rep1	bs-dmso-rep2	ont
chr2:71411474-71417501--:L1HS:REF	98,6%	98,8%	98,3%	0,0%
chr4:146304143-146304145:+:L1HS:NONREF	92,3%	90,4%	90,6%	5,9%
chr7:67126183-67126185--:L1HS:NONREF	86,1%	85,7%	84,7%	9,7%

Reviewer Table 1 – Comparison of replicate *bs-ATLAS-seq* assays for L1 elements with discordant methylation levels between *bs-ATLAS-seq* and Oxford Nanopore Technology sequencing in HCT-116 cells. *Bs-panel*: *bs-ATLAS-seq* result obtained with the panel of 12 cell lines; *bs-dmso-rep1* and *bs-dmso-rep2*: *bs-ATLAS-seq* results obtained for HCT-116 cells treated with DMSO for 2 independent biological replicates; *ont*: nanopore sequencing.

Then, we considered potential inherent differences between bs-ATLAS-seq and ONT. Both techniques are strand-specific, but they target different strands (L1 positive strand for bs-ATLAS-seq and L1 negative strand for ONT). While it is generally assumed that the two L1 strands have identical methylation levels, it is conceivable that certain loci exhibit unique strand asymmetry. Furthermore, while bs-ATLAS-seq cannot discriminate between 5-methylcytosine (5mC) and 5-hydroxymethylcytosine (5hmC), ONT measures 5mC specifically. It is possible that these three loci are hydroxymethylated. As we have no definitive explanation for these rare discordant events between bs-ATLAS-seq and ONT, we prefer not to speculate.

1.9. Methylation levels across the L1 sequence described on page 8, lines 11-14, were previously described in human patient samples by Ewing et al. 2020 (ref 110). That publication should be acknowledged here.

We thank the Reviewer for pointed this oversight. We cited this article elsewhere, but we now also refer to this publication in this context (now ref. 47, p. 9):

(...) a situation previously observed for transgenic CpG islands in reporter genes, as well as at specific L1 loci^{47,119}, and termed "sloping shores"⁵⁹.

Reviewer #2:

Lanciano and colleagues explore the relationship between L1 promoter methylation and transcription factor (TF) binding in a substantial panel of cultured human cells. It is very interesting how these parameters, alongside genomic context, combine to regulate L1 transcriptional activity. The standard of the presented analyses is exceptionally high. I commend the authors, and particularly whoever assembled the many figure panels showing L1 analyses, for the quality of their work. One caveat, emphasised by the variation in L1 regulatory "rules" amongst cultured cells, is that some of the findings may not carry over to other cell types or to in vivo biology, but there L1 regulation is also much more difficult to study. Overall it is a thoughtful and important manuscript that tells us much about L1 regulation and was a pleasure to consider.

We thank the Reviewer for his enthusiasm and helpful suggestions.

2.1. The analysis of YY1 binding and L1 methylation in Fig. 5 does a nice job of showing that the YY1 motif is associated with methylation, in agreement with PMID: 31230816. This agreement could be stated more clearly, even if YY1 binding to that motif is associated with hypomethylation. It is very difficult to untangle that second relationship, because YY1 is both a repressor and an activator (Yin Yang 1), depending on context, and will bind unmethylated DNA more effectively than methylated DNA. TF binding is also dynamic. It remains possible that YY1 binding influences methylation and then, once methylated, no longer binds to a given locus, and this could be noted as well if the authors agree it a reasonable point.

The suggestion of the Reviewer – also made by Reviewer #1 - is fully justified. We now address this possibility in more detail below (please also see Answer #1.3).

I would not propose additional experiments, but to strengthen this section the authors could:

- Present the analysis for Fig. 5E in the same style as Fig. S5C, splitting the YY1 binding +/- into YY1 motif presence +/-, showing whether the YY1 motif is associated with methylation in other cell types.
- Present the analysis for Fig. 5E, except splitting the data instead into intragenic / intergenic elements. The reason for this is that almost all of the elements studied in PMID: 31230816 were deliberately chosen for their position in intergenic regions, away from the influence of genic methylation (which the authors have shown here can make a big difference to L1 methylation). It seems the "rules" are different in YY1 and 2102Ep, and that may be borne out by doing this intragenic / intergenic analysis.
- The H1 and 2102Ep methylomes are very different. In H1, it does not seem that YY1 binds L1 extensively (Fig. 5B) and there isn't H3K4me3 data available to test whether this binding results in higher transcription or not. The opposite is true in 2102Ep, where the data are much more clear. In HEK293 cells (Fig S5B) the YY1 bound young L1s are significantly MORE methylated than the unbound L1s. I think it would be fair to say that the rules around YY1 binding and L1 methylation are not easy to generally define without perturbing the system. page 9, line 44, refers to "embryonic stem cells or embryonal carcinoma cells" yet these relationships are only really clear for 2102Ep. Perhaps then the authors could amend this text to "embryonal carcinoma cells", where the data are clearest, and make a note of future work being required to dissect the YY1 / L1 methylation relationship.

(note: my lab published a previous study evaluating the relationship between the L1HS YY1 motif and methylation PMID: 31230816 and a reader should know that when considering whether these points are reasonable)

We thank the Reviewer for his suggestions, which confirmed and extended the observation made in Sanchez-Luque *et al.* (2019) that among L1 elements not bound by YY1, those lacking the YY1 binding site are often less methylated than those containing a YY1 binding site (**updated Figure S5C**), in agreement with the scenario proposed by the Reviewer. Further research is needed to elucidate the factors that would trigger YY1 release from methylated L1, especially considering our analysis of YY1 ChIP-seq experiments in H1 cells, which suggests that L1 methylation alone is not sufficient for YY1 release (new **Figure S5D**). Given this uncertainty, we have refrained from speculating on this aspect of the model in the revised text (p. 11):

We confirmed that the same locus (chr13Δ31_{L1}), as well as other L1 elements carrying a similar deletion analyzed in the same study, are hypomethylated in H1 and 2102Ep embryonic stem cells and also in HepG2 cells (Figure 5E, blue labeled data points). Therefore, we analyzed separately L1 elements lacking the YY1 motif, those possessing the motif but being unbound by YY1, and those with the motif and being bound (Figure S5C). Interestingly, among unbound L1 elements, those lacking the YY1 motif are less methylated than those with the YY1 motif, in H1, HepG2 or HEK-293 cells (ΔmCG=21%, ΔmCG=17% and, ΔmCG=11%, respectively). In 2102Ep cells, YY1-bound L1 elements exhibit the active chromatin mark H3K4me3 and higher expression levels compared to unbound elements (Figure 5D, Figure S5E and Figure S5F). In contrast to 2102Ep embryonal carcinoma cells, the majority of YY1-bound L1HS and L1PA2 are hypermethylated in the embryonic stem cells H1, and only a minor fraction is hypomethylated and marked by H3K4me3 (Figure S5D and Figure S5F). Considering that 2102Ep cells are nullipotent and blocked in an undifferentiated embryo-like state⁷³, and that H1 cells under our growth conditions are likely in a primed state (see above), our observations are thus consistent with a model whereby YY1 preferentially binds unmethylated L1 loci in naïve pluripotent cells, thereby enabling accurate L1 transcription initiation,

but subsequently mediates L1 de novo DNA methylation upon cellular differentiation during development¹¹⁹, eventually associated with a loss of YY1 binding. Alternatively, alterations of the YY1 motif may incidentally affect other pathways targeting the same sequence and leading to L1 hypomethylation¹¹⁹.

Updated Figure S5C and new Figure 5D:

(C) DNA methylation levels of L1HS elements with (+) or without (-) YY1 binding motifs in their 5' UTR, and actually bound (+) or not (-) by YY1 in embryonal cell lines (H1 and 2102Ep) and other cell lines for which matched YY1 ChIPseq were also publicly available (K-562, HCT116, HepG2, HEK-293). The number of L1HS copies in each subset (n) is indicated at the bottom of the plot. The data are identical to those presented in Figure 5E, but the loci unbound by YY1 were divided into two categories based on the presence or absence of the YY1 motif.

(D) Heatmap displaying L1 methylation (bs-ATLAS-seq), as well as YY1 and H3K4me3 binding (ChIP-seq), at the 5' junction (-1 to +0.5 kb) of L1HS and L1PA2 elements in H1 cells. Loci are sorted by increasing levels of L1 methylation. ChIP-seq signal represents the number of normalized reads per 10-bp bin.

We hope that the revised discussion on the connections between YY1 and L1 methylation now better integrates our work with previous findings.

We also reanalyzed the data shown in **Figure S5C**, separating L1s into intragenic and intergenic elements. The results of this analysis are presented in the **Reviewer Figure 2** below. Since this analysis did not provide additional insights beyond those brought by the updated **Figure S5C**, we have decided not to include it in the revised manuscript.

Reviewer Figure 2:

Reviewer Figure 2 - DNA methylation levels for intra- (green) vs intergenic (white) L1HS loci, and actually bound (+) or not (-) by YY1 in embryonal cell lines (H1 and 2102Ep) and other cell lines for which matched YY1 ChIP-seq were also publicly available (K-562, HCT-116, HepG2). HEK-293 have been omitted since there is no YY1-bound L1 in these cells (see Figure S5C).

Minor Points:

2.2. Line 6, page 6 (authors' page numbering): "the youngest L1HS family appears hypermethylated" could say this is in agreement with a prior analysis of human tissues (PMID: 33186547, Figure S2A).

We have now underlined the agreement with PMID 33186547 (Ref. 47), but instead in the new paragraph introducing the DNA methylation array results to maintain the flow of the text (p. 7):

Remarkably, consistent with bs-ATLAS-seq results, and in agreement with prior nanopore analysis of a couple of human tissues⁴⁷, we observed that the methylation of young L1PAs is high, and generally similar to or higher than that of old L1PAs in most situations.

2.3. Line 24, page 9: please state the absolute median % mC differences for H1 and 2102Ep between YY1-bound and YY1-unbound elements.

This piece of information has now been added to emphasize the difference between these two cell types (p. 11):

In [2102Ep and H1] cells, YY1-bound L1HS elements are significantly less methylated than their YY1-unbound counterparts, although the absolute difference of methylation between the bound and unbound elements is much more prominent in 2102Ep cells ($\Delta\%mCG=31\%$) as compared to H1 cells ($\Delta\%mCG=2\%$) (Figure 5D and Figure 5E).

2.4. Fig. 7. The authors should either perform H3K4me3 ChIP-seq on these HCT-116 cells to measure L1 activation more comprehensively OR note a caveat that the analysis may have overlooked transcriptionally induced L1s that do not generate 3' read-through transcription.

The Reviewer is absolutely correct that some expressed L1 could escape detection through the analysis of 3' readthrough transcription. To address this concern, we intentionally employed a dual approach that combines 3' readthrough analysis with an alternative method for measuring L1 expression using L1EM software, with similar conclusion (**Figure 7E**). As the L1EM algorithm relies on internal reads, we believe that our conclusion that L1 hypomethylation is not sufficient for expression in HCT-116 cells is robust. Although the use of this dual strategy was indicated in the figure itself, it was not detailed in the text. This is now more clearly stated (p. 13):

To exclude unpredictable effects of gene body demethylation on intragenic L1 expression, we analyzed separately L1 loci located outside genes, within genes that are expressed, and within genes that are not expressed, using a dual strategy combining 3' readthrough detection and L1EM (Figure 7E).

2.5. Fig. 6B. The numbers of L1HS copies displayed suggests these are not full-length. If so, how are their methylation levels assessed?

We apologize for the confusion. The numbers correspond well to full-length L1HS elements but considering all cell lines pooled together. Thus, each locus is counted several times but not necessarily in the same methylation category depending on the cell line. We now clarified this point in the legend of **Figure 6B**:

(B) Barplots indicating the absolute number of L1HS elements with low ($mCG \leq 25\%$), medium ($25\% < mCG < 75\%$) or high ($mCG \geq 75\%$) methylation levels obtained by bs-ATLAS-seq in all cell lines combined, and detected as unexpressed (white) or expressed (light green) by the combined 3' readthrough (3' RT) and L1EM⁹³ approaches.

2.6. Line 23, page 7: the authors may note that sloping shores extending into genomic flanks have previously been identified at specific L1 examples (e.g. PMID: 31230816, Fig. 3B), even though I agree that this has not been done systematically to my knowledge.

Thank you for pointing out this oversight. We now added there a reference to Ewing *et al.* (2020) [Ref. 47], as well as to Sanchez-Luque *et al.* (2019) [Ref. 119] (p. 9):

(...) a situation previously observed for transgenic CpG islands in reporter genes, as well as at specific L1 loci^{47,119}, and termed "sloping shores"⁵⁹.

Similarly, we added the references in the discussion (p. 15):

(...) a property previously reported for a GFP reporter gene containing a CpG island and mobilized by an engineered mouse retrotransposon⁵⁹, and for specific L1 loci^{47,119}.

2.7. The Methods state how the YY1 peaks were defined for 2102Ep, but is this bioinformatic approach also applied to the YY1 ChIP-seq data from ENCODE? How clear are these YY1 peaks in H1? It seems like there is relatively little enrichment for YY1 binding to L1HS in H1 cells in general (Fig. 5B) and, although there are beautiful peaks shown for YY1 in the 2102Ep data (e.g. in Fig. S5E) there aren't any examples shown like this for H1. Could one be provided in supplement?

Thank you for pointing out this ambiguity. YY1 ChIP-seq data were analyzed similarly for 2102Ep and H1 cells. This is now explicitly stated in the Methods section (p. 62):

An identical procedure was applied to process our own experimental ChIP-seq data and to reanalyze publicly available datasets (ESR1 in MCF-7 and YY1 in H1 cells). For the other publicly available transcription factor datasets, peaks were computed by and obtained from the Unibind database.

We would like to clarify that **Figure 5B** shows enrichment for transcription factors binding to unmethylated L1s compared to methylated L1s, rather than overall enrichment for YY1 binding to L1s in general. While **Figure 5B** does not show enrichment for YY1 binding to hypomethylated L1s, **Figure 5E** indicates that approximately 40% of L1HS are bound by YY1 in H1 cells, suggesting that YY1 does bind to methylated L1s in these cells. To further clarify this point, we have generated a new **Figure S5D** that displays YY1 binding on individual L1HS and L1PA2 loci, along with L1 DNA methylation and H3K4me3 binding. Additionally, as requested by the Reviewer, we have updated **Figure S5E** to showcase an L1 locus bound by YY1 in both 2102Ep and H1 cells, along with their corresponding RNAseq and H3K4me3 profiles. These changes have been reflected in the main text (p. 11). We believe that these revisions provide a more comprehensive and clearer picture of YY1 binding patterns to L1 elements in H1 cells:

In contrast to 2102Ep embryonal carcinoma cells, the majority of YY1-bound L1HS and L1PA2 are hypermethylated in the embryonic stem cells H1, and only a minor fraction is hypomethylated and marked by H3K4me3 (Figure S5D and Figure S5F).

New Figure S5D and updated Figure S5F:

(D) Heatmap displaying L1 methylation (bs-ATLAS-seq), as well as YY1 and H3K4me3 binding (ChIP-seq), at the 5' junction (-1 to +0.5 kb) of L1HS and L1PA2 elements in H1 cells. Loci are sorted by increasing levels of L1 methylation. ChIP-seq signal represents the number of normalized reads per 10-bp bin.

(F) Genome browser view of two example L1HS loci with distinct promoter DNA methylation profiles (bs-ATLAS-seq), integrated with poly(A)⁺ RNA-seq, YY1 and H3K4me3 ChIP-seq data in H1 and 2102Ep cells. Top, locus in chromosome 18, the YY1 signal is close to the background level, the L1HS element is hypermethylated and non-expressed. Both cell lines have similar profiles. Bottom, locus in chromosome 9, a strong YY1 peak is detected in H1 and 2102Ep cells, where the L1HS element is completely unmethylated and robustly expressed.

2.8. Line 19, page 7: this is also consistent with PMID: 31230816.

The reference has been added.

2.9. Fig. 6D, top right. Why is this element considered to be unexpressed? It very much looks like it has a H3K4me3 peak.

The primary reason for classifying this L1 element as unexpressed is the absence of 3' readthrough transcription in the RNA-seq track, further supported by the lack of internal RNA-seq reads. While the H3K4me3 signal could be genuine, it is notable that the signal is very flat and appears to be restricted to the internal L1 sequence. This pattern differs from that observed at loci with unambiguous transcriptional activity, where the H3K4me3 signal extends to the unique sequence upstream of the L1. Even if uniquely mapping reads are used for ChIP-seq analysis, there is a possibility that the internal reads originate from a non-reference L1 copy and are erroneously mapped to this locus. Therefore, based on the available evidence, we believe that the classification of this L1 element as unexpressed is justified.

Reviewer #3

This important study from Lanciano, Philippe and others is an important step towards understanding the role that full-length LINE-1 elements play in altering the epigenetic and transcriptional landscape of cells, the variation in their effects and expression at a given locus, and some of the mechanisms behind these changes. The authors nicely detail an important technique, bs-ATLAS, for deciphering the methylation status of a discrete LINE-1 locus, and use this and multiple orthogonal techniques to characterize 12 human cell lines ranging from embryonic stem cells to "normal" cell lines to cancer cell lines. The data are striking and show obvious patterns of methylation change related to the CpG island in the 5'UTR of the elements, depict the effects of these methylation differences on the proximal genomic locus of the insertion, and importantly decouple methylation from expression, with the importance of the epigenetics at the locus playing a role in L1 expression. The experiments are well thought out, the findings are well explained and not overstated, and the data are important to the field.

We thank the Reviewer for their supportive comments and for appreciating the quality of our study.

Major Comments:

3.1. Overall, the explanation of statistical tests and rationale for their use could use clarification. Specifically, there should be an extensive statistics section within the STAR methods section. Although the statistics are somewhat contained within figure legends, there is not adequate room for discussion of

methods, and much of the paper's findings depend on association between L1 methylation and the surrounding locus.

We agree with the Reviewer that the description of statistics should be more precise. We have carefully reviewed all figure legends to ensure that the specific statistical tests used are included where needed. Additionally, we have developed the reasoning behind the choice of recurrently used statistical tests in the section of the STAR Methods entitled “**Quantification and statistical analysis**” (p. 64):

Statistical analyses were performed in R and are explicitly described in each Figure legend. Unless otherwise stated:

- *For comparing the distribution of methylation levels for groups of L1 elements that differ by a given genomic feature (e.g., in **Figure 3C**, the average methylation levels of individual L1HS copies in expressed genes vs non-expressed genes), we employed the Wilcoxon ranksum test, a non-parametric test that does not assume normality of the data, with each L1 locus considered as an independent observation.*
- *For comparing the proportion of L1s with different features between two or more independent groups (e.g., in **Figure 3D**, the proportion of L1 in different methylation categories and between different genomic compartments), we used a chi-squared test of independence, a non-parametric test that does not assume normality of the data, with the appropriate contingency table, and without Yates' correction (as the cell counts in the contingency tables were always superior to 5).*

More specific statistical analyses are provided in the relevant Methods sections.

This seeming lack of testing for certain purported enrichments within the data set should be corrected. Examples:

3.2. In section "L1 DNA methylation can be influenced by genic activity" on page 6, line 35: the authors state "In K-562 cells, methylated L1HS elements are enriched in genes". The authors should report statistical values for enrichment and test used in the text or in the figure panel (Figure 3A); moreover the statistics should account for the number of events in intragenic vs. intergenic space, and perhaps use permutations to test the rigor of their association.

The Reviewer is correct that we did not test directly this statement at this stage. We now report the result of a chi-squared test of independence that compares the proportion of methylated L1 inside or outside the genes (p. 8):

*In K-562 cells, methylated L1HS elements are enriched in genes ($p < 0.001$, chi-squared test; **Figure 3A**).*

Furthermore, we updated **Figures 3D**, which compare the proportion of L1s in different categories of methylation between expressed genes, non-expressed genes and extragenic regions, and **Figure 3E**, which compare the proportion of methylated L1s in transcribed vs non-transcribed genomic regions, respectively, by providing the result of chi-squared test of independence, which is detailed in their respective legend.

D L1 methylation by transcriptional activity of the environment **E Methylated L1HS in expressed regions**
(D) Proportion of L1 elements in different methylation level categories according to their genic environment and activity in K-562 (left) or 2102Ep (right) cells. Note that L1PA1 is synonymous with L1HS. N.s.: not significant, **** $p < 0.0001$, chi-squared test.

(E) Distribution of methylated L1HS elements in transcribed (dark green) vs non-transcribed (light green) genomic compartments in K-562 and 2102Ep cells. **** $p < 0.0001$, chi-squared test.

3.3. Although the authors depict an association between the proximal epivariation between L1 alleles in Figure 4, there is no direct test of the effect of the TE itself on the surrounding locus, and the association observed does not have a statistical framework to rely upon. Moreover, on page 8, line 15, the authors state that "the influence of an L1 element on the flanking region could only be assessed if the methylation state of the L1 promoter differs from that of its cognate empty allele". This would appear to bias the results to only examine the 37/78 that show a difference in between the L1 insertion and the empty site. Perhaps the retrotransposition itself affected the methylation status upstream. Perhaps the other 41 loci are indicative of this being a pseudo random event. Statistics and a clearly defined question are important to draw more generalizable results. Again, controlling for the background genomic noise is important, so permutations could be helpful.

We thank the Reviewer for this important point and great suggestion. This was incredibly helpful, and the improvements generated from it substantially strengthen the paper. We have entirely reconsidered the way to analyze the influence of an L1 element on its surrounding genomic region. We clearly separate two distinct hypotheses put forward by the Reviewer that L1 presence (hypothesis #1) or L1 methylation status (hypothesis #2) can influence the epigenetic state of its upstream sequence. We searched for a statistical testing strategy that would take into account the distance to the L1 element, as we typically see a gradient of methylation or demethylation starting from the L1 element. A possible permutation strategy was to scramble L1s and their flanking sequences. However, this would have led to compare methylation states between genomic regions or cellular states that differ enormously or to split the datasets to separate cell types with an inevitable loss of statistical power. Instead, we decided to compare CpG sites proximal or distal to a given L1, which maintains the link between a given L1 and its broader genomic region and with its cellular environment, and takes into account the gradient property of the epigenetic alterations. This statistical framework is now illustrated in the new **Figure 4B**.

New Figure 4B:

(B) Theoretical methylation scenarios and how they can be interpreted to test L1 influence on the upstream flanking region. Under hypothesis #1 (blue), the presence of an L1 element influences the proximal DNA methylation of the upstream genomic region, irrespective of its own methylation profile. Under hypothesis #2 (red), the methylation profile of an L1 element is propagated to its proximal upstream flanking sequence. These scenarios were tested by comparing loci with heterozygous L1 insertions, and thus possessing an empty allele (denoted e) and a filled allele (denoted f). The location of the proximal and distal CpG sites used to evaluate the influence of L1 is indicated in pink (dotted frames on top and intervals at the bottom of the scheme). The proximal CpG site is the closest CpG upstream of L1 in a 300 bp window. The distal CpG site is the closest CpG to L1 1 to 2 kb upstream.

The statistical approach is also extensively described in a dedicated Methods section entitled “**Detection of L1-mediated epivariation**” (p. 63-64):

To determine whether L1 presence or its methylation status influences flanking sequence methylation, we compared the proportion of differentially methylated alleles at proximal and distal CpG sites between filled and empty alleles. We measured proximal methylation at the closest CpG in a 300 bp window upstream of L1, and distal methylation at the closest CpG 1-2 kb upstream of L1. In this approach, the empty allele is used as a proxy for the pre-insertion allele. Although we cannot completely exclude allele-specific methylation independent of L1, this possibility is mitigated by comparing proximal vs distal methylation from the same allele, which accounts for the gradual influence of L1 on its flank. This approach also maintains the association between a given locus and the general epigenetic state of the cell line from which it was extracted. We restricted our analysis to heterozygous L1 loci with at least 5 reads covering each allele and used a Wilcoxon test to compute the difference in methylation between the same CpG site in the empty and filled alleles, considering each read as an observation and yielding to a corresponding p-value.

The first hypothesis tested was that the presence of an L1 element impacts the methylation of its proximal upstream sequence. We classified each CpG site as “influenced” if the methylation of the CpG site in question in the filled allele was statistically different from that of the empty allele

(Wilcoxon test, $p < 0.05$) and "not influenced" if not. Under the null hypothesis, the proportion of CpG sites labelled as "influenced" or "not influenced" is similar at proximal and distal sites. The null hypothesis is consistent with L1 presence on its own having no particular effect on the methylation status of the locus in which it is inserted. To test this hypothesis, we used a chi-squared test of independence.

The second hypothesis tested was that the methylation status of the L1 element influences the methylation of its proximal upstream sequence. Here, we categorized a CpG site as "influenced" if the methylation of the CpG site in question in the filled allele was statistically different from that in the empty allele (Wilcoxon test, $p < 0.05$) and if the methylation level of the L1 promoter (average CpG methylation in the first 200 bp of the L1 sequence) differed by more than 30% as compared to the methylation level of the CpG site in question in the empty allele. Conversely, a CpG site was labeled as "not influenced" if the methylation of the CpG site in question in the filled allele was not statistically different from that of the empty allele but with the methylation level of the L1 promoter still different by more than 30% as compared to the methylation level of the CpG site in question in the empty allele. Any CpG site not falling into one of the above categories was considered inconclusive, and labeled as such. Under the null hypothesis, the proportion of the 3 categories is similar at proximal and distal CpG sites. The null hypothesis is consistent with L1 methylation having no effect on the methylation state of the nearby sequence. The alternative hypothesis is that L1 epigenetic state can mediate local epivariation. To test this second hypothesis, we also used a chi-squared test of independence.

The results confirmed that L1 epigenetic status influences the proximal region upstream of the element. By clearly defining the hypotheses tested, we could also integrate the "inconclusive" methylation scenario to the test, and dismiss pseudo-random events (p. 9-10):

Next, we systematically examined potential L1HS-driven allelic epivariation, considering two distinct hypothetical mechanisms: either the presence of L1, regardless of its methylation state, or the methylation state of L1, could influence the methylation of the upstream flanking region. Among the heterozygous loci, 87 have an upstream CpG within a 300 bp window and were further considered (**Figure 4C**). To test the first hypothesis (**Figure 4B**, blue), we compared the methylation levels of the empty and filled alleles at proximal and distal CpG sites upstream of the L1 insertion (**Figure 4B**, red dotted frames), considering each read as an observation and using a Wilcoxon rank-sum test. While proximal CpG sites provide information about the possible influence of L1, distal CpG sites represent matched genomic background noise. Twentytwo proximal CpG sites out of 87 (25%) were differentially methylated between the empty and filled loci, but this proportion was not significantly different from that observed for the distal CpG sites ($p=0.056$, chi-squared test; **Figure 4E**, left). To test the second hypothesis (**Figure 4B**, red), we categorized each locus as "influenced" if the upstream flanking region is differentially methylated between the empty and filled allele as described above, and if the average methylation level of the L1 promoter differs by more than 30% from the empty allele. Conversely, a locus was considered as "not influenced" when the upstream flanking region is not differentially methylated between the empty and filled allele, but the average methylation level of the L1 promoter differs from that of the empty allele by more than 30%. Finally, the remaining situations were labelled as "inconclusive", and include for instance methylated L1 inserted into a methylated region or unmethylated L1 inserted into an unmethylated region. By comparing the proportion of loci falling in each of these categories, we found that the methylation of the proximal CpG sites is significantly more frequently influenced by L1 methylation than the methylation of the distal CpG sites ($p=0.025$, chi-squared test; **Figure 4E**, right).

New Figure 4C and Figure 4D:

(C) Characteristics of the loci profiled by nanopore sequencing. The y-axis represents the number of loci characterized in the 4 cell lines as filled, heterozygous and with at least one CpG site in a 300 bp upstream window. Ultimately, 87 loci were considered for the subsequent analyses.

(D) Proportion of the observed methylation scenarios at proximal and distal upstream CpGs for the hypotheses described in (B). The relation between the CpG site analyzed and the scenarios observed was assessed by performing a chi-squared test of independence. For hypothesis #1, there is no significant relationship ($\chi^2(df=1, N=174)=3.66, p=0.056$), but there is a significant relationship for hypothesis #2 ($\chi^2(df=2, N=174)=7.35, p=0.025$), indicating that L1 methylation state mediates proximal epivariation.

Minor comments:

3.4. The embryonal carcinoma 2102E cell line has a distinct L1 hypomethylation pattern when compared to K-562; although the pattern is likely due to the poised/early developmental state, this finding lacks a clear explanation in the paper. A clearer breakdown of why H1 and 2102E are different than the other lines would be helpful for a broader audience.

Thank you for the suggestion. We now describe the specificities of 2102Ep and H1 cells (p. 11):

(...) 2102Ep cells are nullipotent and blocked in an undifferentiated embryo-like state⁷³, and that H1 cells under our growth conditions are likely in a primed state (...).

Please note that we also present an entirely new set of results revealing the distinctive profile of young L1 families in contexts associated with early embryogenesis, extra-embryonic tissues, and male germ line tumors (new **Figure 2**, and answer #1 to Reviewer #1), which is relevant to this question.

3.5. For active elements (Figure 6C) what publications are you using to define this activity? This was not apparent in the manuscript, nor in the STAR methods.

We apologize for the omission. The information was indeed included in one of the supplementary Table (now **Table S8**). We have added this information in the legend of **Figure 6C**:

(C) Barplots indicating the absolute number of expressed L1HS elements across the different cell lines, for non-intact (white) and intact copies (dark green). Among the expressed L1 elements, the associated pie charts show the proportion of copies with published evidence of retrotransposition competence (detailed in **Table S8**).

We also added a paragraph in the Methods section (p. 61):

*We further annotated the expressed elements for intactness (i.e., without stop codon in ORF1 or ORF2 according to the L1Base2 database ¹⁹⁴) and for published evidence of retrotransposition activity, as determined by cell culture assays or by the identification of transduction events deriving from the locus. The corresponding publications are listed in **Table S3** for all L1 elements detected by bs-ATLAS-seq across all cell lines, and in **Table S8** for the subset of expressed L1 elements.*

Referees' report, second round of review

Reviewer #1: The authors have thoughtfully and thoroughly addressed my points. The new analyses, figure panels, and additions/modifications to the text in response to comments from all 3 reviewers have, in my opinion, substantially enhanced the work. I have no further concerns.

Reviewer #2: The authors have addressed all of my comments in full and I have no reservations in recommending publication.

Geoff Faulkner (University of Queensland)

Reviewer #3: I appreciate the authors' diligent response to the myriad comments and suggestions, and think they have done a great job explaining and extending the statistical analyses. I have no further comments.

Authors' response to the second round of review

NA